# Comprehensive characterization of IFNγ signaling in acute myeloid leukemia reveals prognostic and therapeutic strategies

Bofei Wang[1,11], Patrick K. Reville [1,11], Mhd Yousuf Yassouf [1], Fatima Z. Jelloul [2], Christopher Ly [1], Poonam N. Desai[1,3], Zhe Wang[1], Pamella Borges[1,4], Ivo Veletic[1], Enes Dasdemir [1,4], Jared K. Burks [1], Guilin Tang [2], Shengnan Guo[5], Araceli Isabella Garza[1], Cedric Nasnas [1], Nicole R. Vaughn[1], Natalia Baran [1], Qing Deng[6], Jairo Matthews[1], Preethi H. Gunaratne[4], Dinler A. Antunes[4], Suhendan Ekmekcioglu [7], Koji Sasaki [1], Miriam B. Garcia[8], Branko Cuglievan[8], Dapeng Hao [5], Naval Daver [1], Michael R. Green [6,9], Marina Konopleva [1,10], Andrew Futreal [9], Sean M. Post[1] & Hussein A. Abbas [1,9] ✉

Interferon gamma (IFNγ) is a critical cytokine known for its diverse roles in immune regulation, inflammation, and tumor surveillance. However, while IFNγ levels were elevated in sera of most newly diagnosed acute myeloid leukemia (AML) patients, its complex interplay in AML remains insufficiently understood. We aim to characterize these complex interactions through comprehensive bulk and single-cell approaches in bone marrow of newly diagnosed AML patients. We identify monocytic AML as having a unique microenvironment characterized by IFNγ producing T and NK cells, high IFNγ signaling, and immunosuppressive features. IFNγ signaling score strongly correlates with venetoclax resistance in primary AML patient cells. Additionally, IFNγ treatment of primary AML patient cells increased venetoclax resistance. Lastly, a parsimonious 47-gene IFNγ score demonstrates robust prognostic value. In summary, our findings suggest that inhibiting IFNγ is a potential treatment strategy to overcoming venetoclax resistance and immune evasion in AML patients.

Acute myeloid leukemia (AML) is a clonal disorder characterized by the presence of immature blasts and arrested differentiation of malignant myeloid blasts in the bone marrow[1]. Discoveries of the genetic underpinning of AML provided better understanding of the biology, prognosis, and treatment of this disease[2–4]. This has led to the recent FDA approval of several targeted therapies that are improving the care of patients with AML[5], including the targeted BCL2 inhibitor, venetoclax, which has been transformative in treating older patients with AML when used in combination with azacitidine[6] or low dose cytarabine[7], and emerging evidence of its impact in combination with

intensive chemotherapy in younger patients[8,9]. Venetoclax combined with azacitidine leads to a response in almost two-thirds of AML patients; however, the remaining third have primary resistance or experience early relapse[6]. The mechanisms of resistance to venetoclax are not fully understood but monocytic subclones are suggested to have inherent resistance to venetoclax[10,11].

The relationship of the bone marrow immune microenvironment and the pathogenesis of AML is also incompletely understood. Dysregulated inflammatory pathways have been implicated in leukemogenesis and the maintenance of leukemic blasts[12,13]. On the one hand,

AML-intrinsic repression of inflammation via *IRF2BP2* may contribute to AML cell survival[14]. Other findings suggest that pro-inflammatory cytokines secreted in the microenvironment can induce leukemogenesis and impact the proliferative capacity of hematopoietic cells. For instance, tumor necrosis factor alpha (TNFα) induces proliferation in myeloid neoplastic cells[15]. Also, interleukin-1 (IL-1) is a druggable target that stimulates the production of other inflammatory molecules promoting AML progression[16,17].

Type 1 interferons, interferon alpha (IFNα), beta (IFNβ), and omega (IFNω), and the type 2 interferon, gamma (IFNγ), are major players in the inflammatory response to cancer[18]. While most cells, including monocytes, macrophages, and other immune cells, produce type 1 IFNs, IFNγ is primarily produced by T and natural killer (NK) cells[19]. IFNα treatment is FDA-approved for myeloproliferative neoplasms (MPN) and tyrosine-kinase resistant chronic myeloid leukemia (CML) and has also been used in the treatment of minimal-residual disease (MRD) in favorable-risk AML, with evidence of conversion to an MRD-negative state in some patients and improvements in relapse-free survival[20–22]. Additionally, IFNγ treatment has been proposed for post-transplant AML patients to induce the expression of human leukocyte antigen (HLA) class II and restore graft versus leukemic immune responses[23]. On the other hand, recent findings have suggested that chronic IFNγ promotes cancer growth, abrogates T cell cytolytic activity, and drives CD8 cells towards an exhausted phenotype[24,25]. Additional pro-tumorigenic effects of IFNγ include the induction of immune checkpoint receptors[26], enhancement of tumor metastasis[27], and promotion of hyper-progression after immunotherapy[28]. These data underscore the dichotomous nature of IFNγ signaling in both the pathogenesis of cancer and immunotherapy response, emphasizing the need for further investigation of its role in leukemia.

In this work, we use a complement of bulk and single-cell approaches in samples from patients newly diagnosed with AML to disentangle the relationship between AML blasts and their immune microenvironment. We show that IFNγ signaling in monocytic and del7/7q subtypes contributes to a unique immune microenvironment. Further, we leverage functional dependency datasets and ex vivo experimentation on primary patient samples to elucidate potential IFNγ-associated dependencies and mechanisms of venetoclax resistance unique to AML. Our work provides a deep characterization of IFNγ signaling and its association with venetoclax resistance in AML.

## Results

### Differential IFNγ activity in distinct subgroups of AML patients

We used single-sample gene set enrichment analysis (ssGSEA) to examine the transcriptional programs linked to IFNγ signaling in 672 newly diagnosed adult AML patients from 3 independent datasets: TCGA[4], MD Anderson Cancer Center (MDACC)[29], and BEAT-AML[3]. Patient characteristics were summarized in Supplementary Data 1. By applying ssGSEA to each sample individually, we calculated independent enrichment scores for each gene set-sample pairing. There was a positive correlation between the IFNγ signaling score, and immune activation pathways curated from the Gene Ontology (GO), Hallmark, and Reactome gene set collections of the Molecular Signature Database (MSigDB) (Fig. 1A, Supplementary Data 2). Additionally, IFNγ signaling score had an approximately normal distribution, suggesting a varying extent of signaling among individuals with AML (Supplementary Fig. 1A).

We next examined clinical characteristics associated with IFNγ signaling score and found a negative correlation with blast percentage (r = −0.41, $p < 2.2 \times 10^{-16}$) (Fig. 1B). Additionally, we observed significant differences in IFNγ signaling score across AML differentiation states, as defined by the French American British (FAB) classification (Kruskal-Wallis test, $p = 1.76 \times 10^{-8}$) and cytogenetic groups (Kruskal-Wallis test, $p = 0.01$). Specifically, among AML patients with diploid cytogenetics, those with monocytic differentiation (FAB M4/M5) had the highest

IFNγ signaling scores (Fig. 1C). Patients with diploid cytogenetics but without a reported FAB classification, referred to as diploid, not otherwise specified (NOS), displayed intermediate levels of IFNγ signaling scores, likely indicating that this group contained a combination of monocytic and non-monocytic patients.

Among AML patients with non-diploid cytogenetics, the IFNγ signaling score was found to be highest in core-binding-factor (CBF) AML with inv(16). This score was significantly higher than that of t(8;21) CBF AML ($p = 5.49 \times 10^{-4}$), which is noteworthy because inv(16) AML typically exhibits a myelomonocytic (M4) differentiation, while t(8;21) CBF leukemia is usually more myeloid (M2) (Fig. 1D)[30]. Among patients with non-CBF, non-diploid AML, those with a deletion in chromosome 7/7q (del7/7q) exhibited highest IFNγ signaling scores (Fig. 1D). Notably, sorted CD34[+] cells which marks the healthy stem cells from 17 healthy donors[3] had markedly lower levels of IFNγ signaling scores than did those from AML patients (Fig. 1E), suggesting that IFNγ pathway signaling is a predominant feature in AML. Further, we assessed IFNγ concentrations in the sera of 43 consecutively collected, newly diagnosed AML patients that present to our center and observed elevated level of IFNγ in sera of 67.4% of patients that exceeded those typical of the healthy reference group, which fortifies the notion that IFNγ is dysregulated at the time of diagnosis in AML patients and underscores the need for further exploration into its potential clinical implications and therapeutic utility (Fig. 1F). Overall, these findings indicate that IFNγ activity varies across AML subgroups and is associated with cell lineage and cytogenetics, both crucial predictors[31,32].

Given that HLA class 1 and 2 are major downstream targets of IFNγ signaling[33,34], we assessed the correlation of IFNγ signaling with HLA class 1 and 2 expression. We found a significant positive correlation between IFNγ signaling score and HLA class 1 and 2 expression (r = 0.56 and 0.52, respectively; $p < 2.2 \times 10^{-16}$) (Fig. 1G, Supplementary Fig. 1B, C). IFNγ signaling score was also associated with T cell dysfunction score, T cell exhaustion score, and T cell senescence score, consistent with the fact that chronic IFNγ can lead to T cell dysfunction[24] (Fig. 1G, Supplementary Fig. 1D–F). This finding suggested that an inflamed immune microenvironment is correlated with IFNγ signaling activity in the AML bone marrow and the presence of T cell dysfunction.

We next employed immune deconvolution with CIBERSORTx[35] to correlate cellular composition with IFNγ signaling activity. Consistent with the role of IFNγ in promoting CD4 T cell activation[36,37], we found a positive correlation between IFNγ pathway with helper and regulatory CD4 T cells (Supplementary Fig. 1G). The cell type most strongly correlated with IFNγ signaling in bulk deconvolution analysis was with monocytes (r = 0.41; $p < 2.2 \times 10^{-16}$) (Fig. 1H). This was consistent with prior findings of the enrichment of inflammatory response signaling with monocytic differentiation[14]. It should be noted that deconvolution analysis cannot differentiate between normal and abnormal monocytic cells. We therefore hypothesized that this correlation with monocytic cells may be related to monocytic differentiation prompting us to examine the respective contributions of AML cellular components to the expression of IFNγ pathway.

### Characterization of IFNγ signaling in newly diagnosed AML bone marrow aspirates using single cell RNA (scRNA)

To address the limitations of bulk RNA profiling in discerning cellular subsets and the relative contributions of cells to IFNγ signaling, we performed scRNA profiling on 20 bone marrow aspirates from AML patients at the time of their diagnosis. An overview of the demographic, clinical, and molecular features of the patient cohort is provided in Fig. 2A and Supplementary Table 1. Briefly, the cohort had a median age of 73 years (range 52–87), with 4/20 (20%) being female and 13/20 (65%) having de novo disease. Seven patients had diploid cytogenetics (3/7 non-monocytic, and 4/7 monocytic), 5 patients each had del7/7q or deletion in chromosome 5/5q (del5/5q), and 3 patients had both del5/5q and del7/7q (double deletion). The most frequent

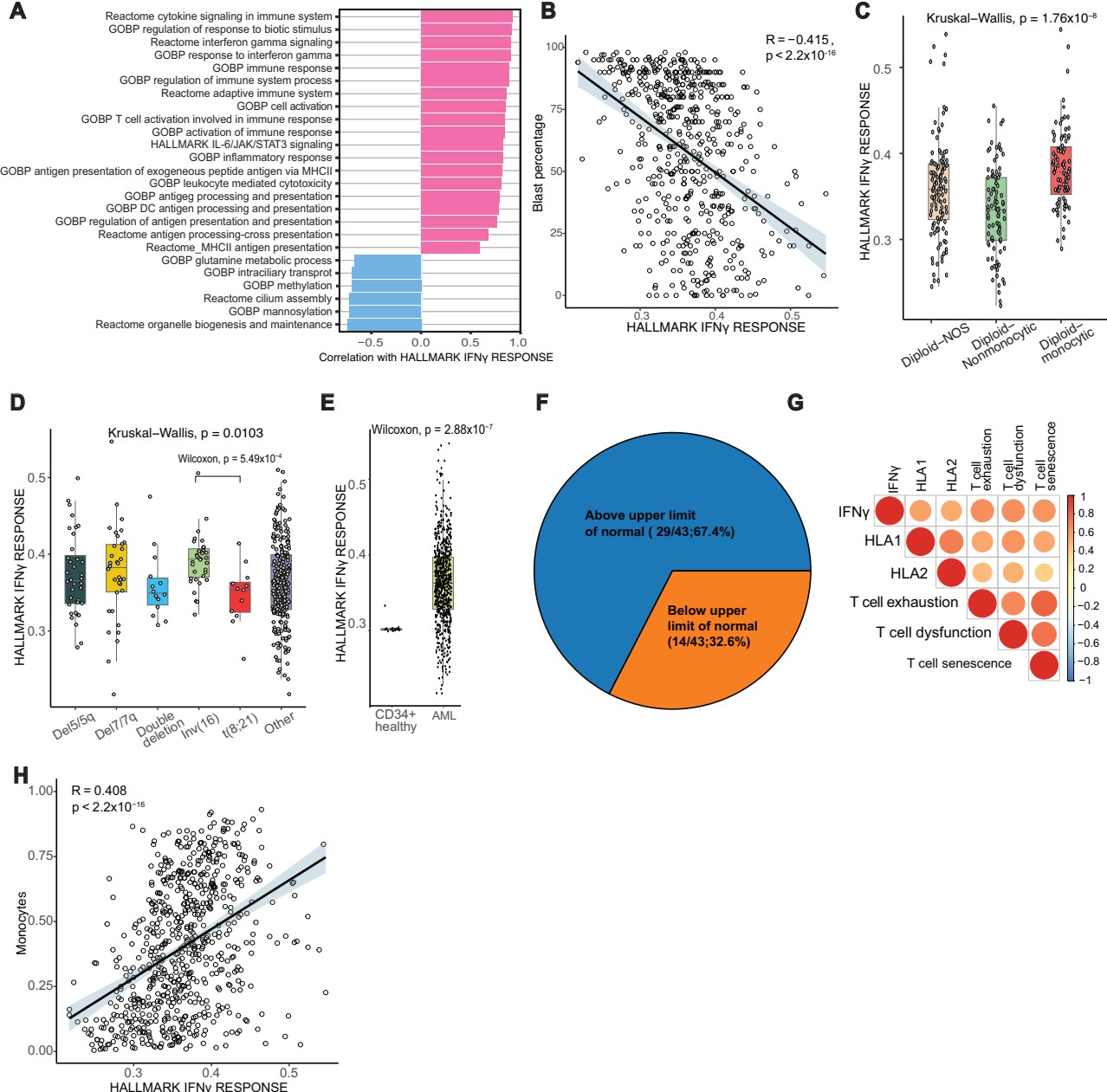

**Fig. 1 | Bulk RNA profiling identifies conserved IFNγ signaling in AML samples.**
**A** Single-sample gene set enrichment analysis of TCGA, MDACC, and BEAT-AML bulk RNA sequencing datasets using pathways curated from Gene Ontology (GO), Hallmark, and Reactome gene set collections of the Molecular Signature Database. **B** Correlation of blast percentage in the bulk RNA sequencing cohorts and Hallmark IFNγ response pathway. Error band represents 95% confidence interval. *T* test was used to evaluate the significance of Pearson correlation. **C** Hallmark IFNγ response score by specific FAB classification in patients with diploid cytogenetics (*n* = 294; NOS = not otherwise specified; Center line represents the median and lower and upper bounds of box correspond to the first and third quartiles). **D** Hallmark IFNγ response score by specific cytogenetic groups (*n* = 378); double deletion indicates patients with both a chromosome 5/5q and 7/7q loss. Center line represents the median and lower and upper bounds of box correspond to the first and third

quartiles. Two-sided Wilcoxon test was used to compare inv(16) with t(8;21).
**E** Hallmark IFNγ response score comparing AML samples and healthy CD34⁺ sorted bone marrow cells. Center line represents the median and lower and upper bounds of box correspond to the first and third quartiles. Two-sided Wilcoxon test was used. **F** Pie chart showing the percentage of newly diagnosed AML patients with elevated IFNγ level compared to the normal range. **G** Correlation of IFNγ response pathway with HLA1, HLA2, T-cell exhaustion, T-cell dysfunction, and T-cell senescence scores (see also Supplementary Fig. 1B–F). **H** Correlation of Hallmark IFNγ response with monocytes as determined through CIBERSORTx immune deconvolution of bulk RNA profiling data. Error band represents 95% confidence interval. *T* test was used to evaluate the significance of Pearson correlation. Source data are provided as a Source Data file.

mutations were *IDH2*, *NPM1*, and *DNMT3A* in 7/20 (35%) and 6/20 (30%) and 6/20 (30%) of patients respectively.

A total of 107,067 cells passed quality control and were further analyzed (see Methods and Supplementary Figs. 2, 3). Nine clusters emerged and were identified by canonical gene expression: AML cells (52.8%), early progenitors (2.6%), T cells (25.5%), B cells (3.5%),

monocytes (3.5%), erythroid cells (5.2%), natural killer (NK) cells (2.2%) and dendritic cells (DC) (1.6%) (Fig. 2B–D, Supplementary Table 2; Supplementary Data 3). Less than 3% of cells lacked distinct marker gene signatures and were classified together with plasma cells and megakaryocytes as "other". The non-AML compartment was mainly composed of T cells (53.97% of cells) and showed variable distribution

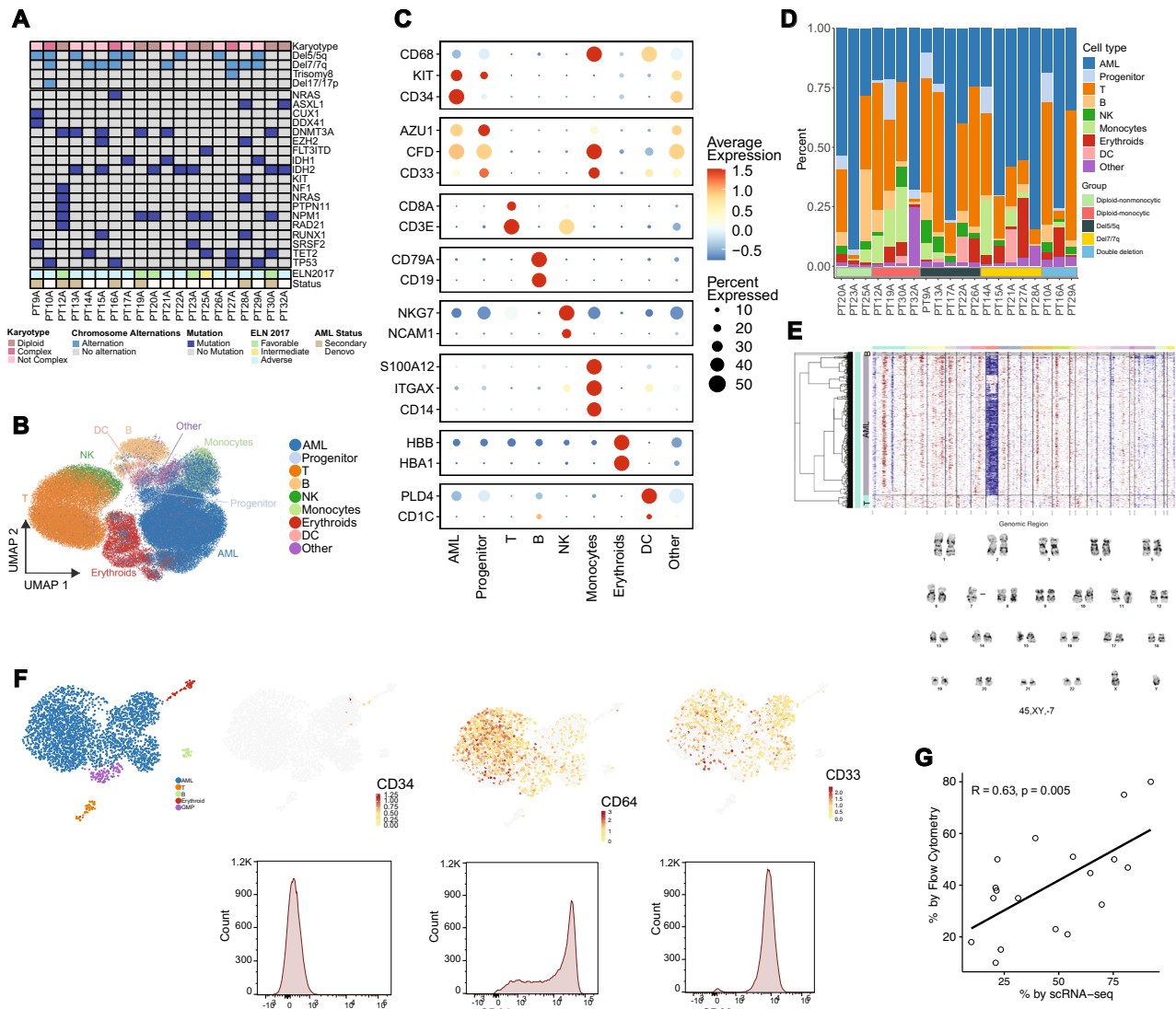

**Fig. 2 | Single cell RNA profiling of newly diagnosed AML bone marrows.**
**A** OncoPrint of patients included in the single cell RNA (scRNA) profiling. **B** UMAP projection of 107,067 cells passing quality control (see Supplementary Fig. 2 for QC) and cell cluster identities. **C** Marker gene expression for key marker genes defining cell cluster identities. **D** Relative proportions of cells by patients.
**E** Example of concordance between inferCNV results and the karyotype of a patient (PT28A), with a loss of chromosome 7 shown in G-banding karyotype and loss of

transcripts corresponding to chromosome 7 in inferCNV map. **F** Example of concordance between gene expression of monocytic AML-defining markers determined by scRNA profiling and protein expression determined by flow cytometry (CD34 negative, CD33 positive, and CD64 positive) in a representative patient with monocytic AML (PT32A). **G** Concordance of AML blast count by flow cytometry and scRNA profiling for all patients profiled. *T* test was used to evaluate the significance of Pearson correlation. Source data are provided as a Source Data file.

across cytogenetic groups and patients, suggesting the intrinsic heterogeneity of the cellular composition in tumor immune microenvironment of AML patients.

To validate the identity of AML cells more specifically, we used infer copy number variation (inferCNV) as described in previous studies[38]. This approach successfully recapitulated the conventional cytogenetic characteristics of patients (Fig. 2E). Additionally, we were able to recapitulate monocytic differentiation in AML using gene expression of relevant marker genes (*CD34*, *CD33*, or *FCGR1A/CD64*) whose protein expression characterization was validated with flow cytometry (Fig. 2F; Supplementary Table 3). The proportions of AML cells identified in scRNA were similar to those determined by histopathologic review and were positively correlated with the frequency of AML cells determined by flow cytometry (r = 0.63, *p* = 0.005) (Fig. 2G). These consistent results indicated that scRNA analysis accurately defined AML subsets and provided a reliable basis for further downstream analysis.

## Disentangling AML and immune cells at the single-cell level reveals IFNγ signaling activation in leukemic blasts

In addition to AML cells, we focused our subsequent analysis on T and NK cells, as the latter cells serve as the primary mediators and effectors for the IFNγ signaling pathway[18]. We first evaluated the effects of IFNγ signaling on the AML cells and tumor microenvironment using AUCell[39]. We scored both AML and T cells to obtain a single cell-level assessment of IFNγ signaling activity. The relative difference in the expression of IFNγ signaling activity in T cells among the cytogenetics groups was smaller than that observed in AML cells (Fig. 3A). Specifically, we observed that AML cells in patients with diploid monocytic AML had the highest expression of IFNγ signaling scores (ratio of AML to T cells = 0.93), while the non-monocytic, diploid AML cells had the lowest (ratio of AML to T cells = 0.56) (Fig. 3A, B, Supplementary Fig. 4A). Among nondiploid cytogenetic groups, IFNγ signaling activation was highest among those with del7/7q (Fig. 3B). Importantly the observation of high IFNγ signaling in monocytic AML was validated in

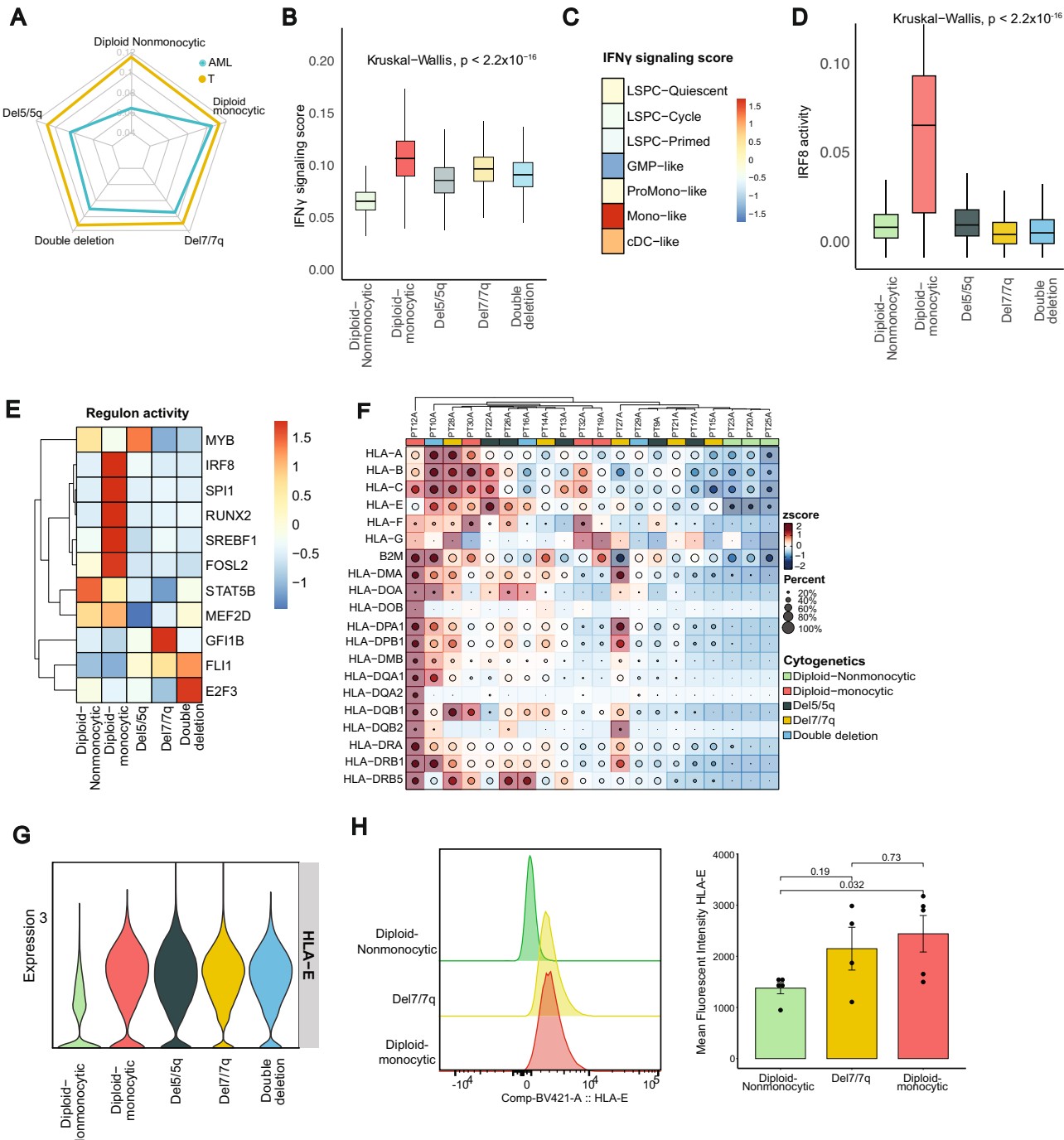

**Fig. 3 | IFNγ signaling in AML blasts is dependent on phenotypic and cytogenetic groups. A** Radar plot of single cell-level assessment of IFNγ signaling scored, generated with AUCell, comparing T cells and AML cells from patients with diploid non-monocytic, diploid monocytic, del5/5q, del7/7q, and double deletion (both del5/5q and del7/7q) AML. **B** IFNγ signaling score across specific AML subgroups (*n* = 56,168 AML cells; Center line represents the median and lower and upper bounds of box correspond to the first and third quartiles). **C** IFNγ signaling score across AML differentiation hierarchies as described in Zeng et al. 2022. **D** Interferon regulator factors 8 regulon activity across AML groups determined by SCENIC (*n* = 5617; Center line represents the median and lower and upper bounds of box correspond to the first and third quartiles). **E** Regulon activities of 11 core transcriptional regulators reported by Eagle K. et al. visualized by AML groups. **F**. Heatmap of HLA class 1 and class 2 expression across patient samples. **G** *HLA-E* RNA expression in AML subtypes. **H** Representative histogram of HLA-E expression in AML blast cells as detected by flow cytometry (left) and quantification of mean fluorescent intensity (MFI) of HLA-E expression between diploid non-monocytic (*n* = 5), del7/7q (*n* = 4), and diploid monocytic (*n* = 5) AML patient samples (right; see also Supplementary Fig. 4F). Data are presented as mean values ± SD. Source data are provided as a Source Data file.

two independent scRNAseq cohorts[40,41] (Supplementary Fig. 4B, C). This suggested that the differences in IFNγ signaling scores noted in bulk RNA data were likely driven by differences within the AML cells themselves.

Zeng et al.[32] recently described a cellular hierarchy of AML leukemic stem cells representing distinct maturation states including LSPC-Quiescent, LSPC-Primed, LSPC-Cycle, GMP-like, ProMono-like, Mono-like and cDC-like. We investigated IFNγ signaling activity across these AML hierarchies and found again that it was highest among cells in the monocyte-like state (Fig. 3C). Zeng et al. also revealed 4 distinct subtypes in bulk cohort based on the composition of their leukemia cell hierarchy. We applied this method on our bulk cohort and validated the IFNγ signaling activity in these patients, and it was highest in the mature state (enriched for mature Mono-like and cDC-like blasts) while lowest in GMP states (Supplementary Fig. 4D). Because IFNγ activity and signaling are epigenetically regulated, we employed SCENIC, a computational tool for inferring transcription factors from constructed gene regulatory networks using scRNA-seq data[39]. Our analysis revealed high regulon activities of interferon regulator factors (IRFs) in AML cells, with elevated levels of *IRF1* and *IRF5* regulons in del7/7q and del5/5q, respectively (Supplementary Fig. 4E). Notably, we also observed an elevated *IRF8, IRF2, IRF3, IRF7* regulon in diploid AML cells with monocytic differentiation (Fig. 3D; Supplementary Fig. 4E), consistent with their role as a lineage-determinant factor promoting monocytic differentiation[42–44]. Of the recently reported 19 core transcriptional regulators of lineage survival in AML[45], 11 were predicted by SCENIC, all of which demonstrated significant differences in regulon activity across cytogenetic groups (Fig. 3E). These observations indicate that the activation levels of IFNγ signaling in AML cells are associated with distinct cellular states and hierarchies, and that disparate regulons of IFNγ signaling characterize various patient subgroups.

The antigen presentation machinery, including HLA expression, are prominent downstream targets of IFNγ pathway activation and critical for immune recognition. In our analysis, the expression of HLA class 1 and 2 genes was also notably different across AML cells from different subgroups, where diploid-nonmonocytic AML patients had the lowest expression of HLA genes (Fig. 3F). To validate these findings, we performed spectral flow cytometry on 14 peripheral blood samples from AML patients at diagnosis representing 5 patients with diploid-nonmonocytic AML, 5 with diploid-monocytic AML, and 4 patients with del7/7q AML (Supplementary Table 4). Interestingly, *HLA-E*, a non-classical class 1 HLA with regulatory functions, was more highly expressed at both the RNA and protein level in diploid-monocytic AML blasts than in diploid nonmonocytic blasts, confirming our scRNA-seq findings (Fig. 3G, H, Supplementary Fig. 4F). However, flow cytometric protein expression of HLA class 2 was highly expressed on most AML samples and was not significantly different across AML subtypes (Supplementary Fig. 4G, H). These findings, taken together, indicates that the monocytic AML microenvironment is characterized by high IFNγ signaling that correlates with high HLA-E expression, possibly representing an immune evasion strategy to limit CD8 and NK cytotoxic activity through inhibitory interactions with *NKG2A*.

### High IFNγ pathway activation in AML cells is correlated with T cell inflamed microenvironment and distinct regulatory pathway activation

To identify the source of IFNγ in the AML microenvironment, we assessed *IFNG* expression in AML and effector immune cell subsets. *IFNG* was most prominently expressed in CD8 and NK cells with a smaller contribution from CD4 T cells and almost no expression of *IFNG* in AML cells (Supplementary Fig. 5A). To comprehensively quantify IFNγ production, we ran gene set enrichment analysis on AML, CD4, CD8, and NK cells with the IFNγ production gene set from GO. Again, we found that CD8 T cells showed the highest production

activity followed by NK cells while the AML cells had the lowest activity (Fig. 4A). To further elucidate the immune microenvironmental differences among cytogenetic and phenotypic AML groups, we assessed the T and NK cell composition within each group. Interestingly, diploid non-monocytic AML had the lowest infiltration of T cells and had the lowest proportion of NK cells with expression of the inhibitory receptor *NKG2A* which binds to *HLA-E* to suppress the cytolytic activity of NK cells (Supplementary Fig. 5B–F)[46].

To further elucidate the cell-cell interactions in the immune microenvironment we employed the CellChat[47] and MultiNicheNet tools[48]. CellChat quantifies interaction strength by aggregating the communication probabilities across all cell group pairs. We observed a significantly higher communication probability between T cell-AML interactions in diploid monocytic AML compared to other groups ($P < 0.0001$) and no predicted T cell-AML interactions in diploid non-monocytic AML (Fig. 4B, top). This observation held when the strength was scaled to account for confounding factors (Fig. 4B, bottom). We then employed MultiNicheNet to predict the top 100 receptor-ligand interactions among AML, CD8 T, CD4 T, and NK cells (Supplementary Data 4). Interestingly, IFNγ from CD8 T cells and NK cells was predicted as a top interaction with AML cells only for patients with diploid monocytic AML (Fig. 4C, Supplementary Fig. 5G–J). Of note, del7/7q AML also showed signs of both a pro- and anti-inflammatory microenvironment with prominent predicted interactions involving TNFα and the TGFβ pathway (Supplementary Fig. 5I). Given these data suggesting an inflamed microenvironment as a prominent feature of diploid monocytic AML, we further analyzed receptor-ligand interactions in the diploid monocytic group only (Supplementary Data 5). This confirmed that IFNγ from CD8 T and NK cells acting on AML cells as a prominent feature in this group of patients (Fig. 4D, E). Interestingly, this analysis also predicted other prominent inflammation-related interactions including TNFα, CCL3, CCL4, and the SIRPα pathway which is an emerging drug target in AML[49] (Fig. 4E). To assess the capacity of AML cells to directly induce INFγ secretion, we performed co-culture assays of AML cells with T cells in primary cells of two AML patients and two AML cell lines, MOLM13 and THP1. Our analysis revealed that in isolation, AML cells did not secrete INFγ, while T cells had a low basal level of INFγ secretion (Fig. 4F). In contrast, when AML cells were co-cultured with T cells, there was a consistent and significant elevation in the levels of INFγ detected in the supernatant at both assessed time points relative to T cells in isolation (Fig. 4F). This increase in INFγ levels upon co-culture suggests that AML-T cell interactions is sufficient for INFγ production. Of note, strong T cell activation with agonistic anti-CD3/CD28 stimulation was associated with a considerably greater increase in INFγ secretion from T cells compared to the co-culture assay (Fig. 4F). This substantial increase suggests that, while direct AML-T cell contact is capable of initiating INFγ production, other factors may contribute to amplifying or dampening T cell activation. To further validate T cell-AML interactions, we employed COMET based multiplex immune-fluorescence (IF) panel to spatially quantify the bone marrow microenvironment in 2 AML bone marrow samples. Using this technology we showed HLA-E⁺CD34⁺ AML cells are in closer proximity to CD3⁺ T cells than that of HLA-E⁻ AML cells, suggesting an immunosuppressive neighborhood between HLA-E expressing AML cells and T cells (Fig. 4G, H; Supplementary Table 5). Taken together, these findings suggest that IFNγ signaling in AML which is likely instigated from neighboring T/NK cells confers immune-evasion via upregulation of HLA-E.

### High IFITM3 expression predicts worse survival in AML

To delineate the effects of IFNγ signaling in AML and its dependency genes, we calculated correlations of IFNγ signaling score with all expressed genes in the single cell data. The top genes positively correlated with IFNγ signaling score included *CD74, IFITM3, and HLA-E* (Fig. 5A, Supplementary Data 6). Consistent with our findings on

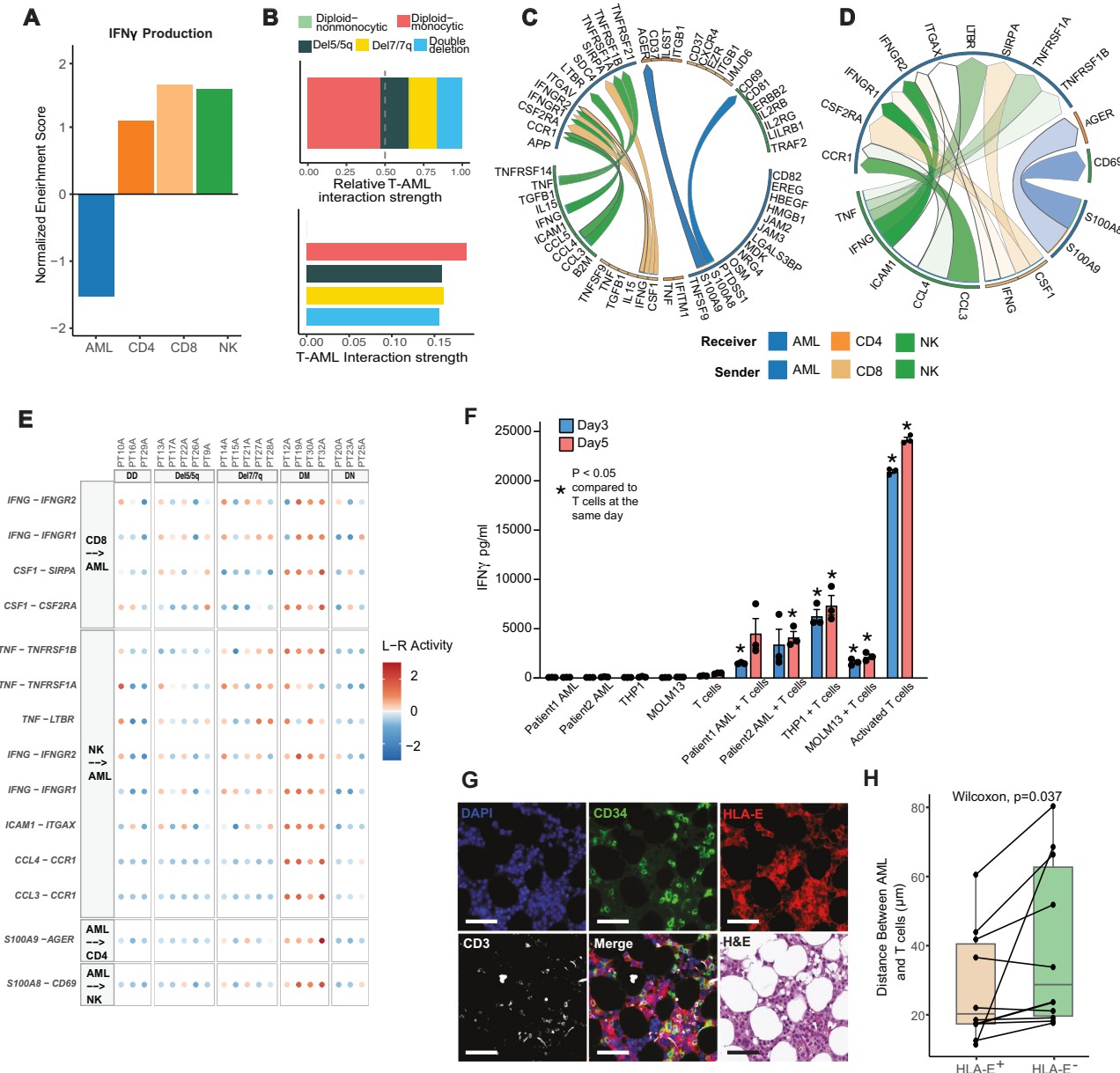

**Fig. 4 | CD8 T and NK cell IFNγ production in the AML bone marrow shapes a unique immunosuppressive microenvironment. A** Normalized enrichment score of GSEA of IFNγ production pathway for AML, CD4, CD8, and NK cells. **B** T cell-AML interactions and interaction strength among AML groups predicted by CellChat. **C** Circos plot of the top predicted non-AML-to-AML ligand-receptor interactions within the diploid monocytic subset among the top 100 ligand-receptor interactions predicted by MultiNicheNet. **D** Circos plot of the top 50 ligand-receptor interactions within the diploid monocytic subset only. **E** Dot plot of top 50 ligand-receptor interactions within the diploid monocytic subset only, excluding AML-to-AML interactions. DD double deletion, DM diploid-monocytic, DN diploid-nonmonocytic. **F** IFNγ levels in culture media were assessed by ELISA after a 72 and 120-h co-culture of CD14+, CD34 + AML blasts, THP1 cells or MOLM13 cells with healthy donor T cells. Control groups included blasts alone, T cells alone, and

T cells co-cultured with DynaBeads T cell activator. Co-culture data is from 1 patient with CD14+ blasts, 1 patient with CD34+ blasts, THP1 cells or MOLM13 cells with each condition having 3 replicates. Data are presented as mean values ± SEM. Two-sided *t* test was used (*$p < 0.01$). **G** Representative multiplex IF panel from 2 patient bone marrow biopsy samples imaged using Lunaphore showing a representative image enriched for CD34+ AML cells, serial section stained with DAPI, CD34, HLA-E, and CD3, H&E, and merged section shown for comparison with scale bar indicating 50 μm. The experiment was repeated 10 times. **H** Boxplot of distance between CD3 T cells and AML cells by whether AML cells express HLA-E or not. ($n = 10$; Center line represents the median and lower and upper bounds of box correspond to the first and third quartiles). Two-sided Wilcoxon test was used. Source data are provided as a Source Data file.

cellular hierarchy (Fig. 3C), markers for GMP cells (*AZU1, CFD*) had strong negative correlation with IFNγ signaling score (Fig. 5A). The average score at a patient level also showed a significant association between IFNγ and *IFITM3* (Fig. 5B). *IFITM3* encodes an IFNγ-induced protein which has been suggested to play a role in tumor progression of multiple cancers including B-ALL, mantle cell lymphoma, colorectal,

prostate, and hepatocellular carcinoma[50,51]. The high correlation of *IFITM3* expression with IFNγ signaling prompted us to evaluate whether IFNγ directly stimulates the expression of *IFITM3* in AML cells. We isolated CD14+ and CD34+ AML blasts from two patients each and performed IFNγ stimulation for 24 h, followed by *IFITM3* protein level measurement via flow cytometry. Indeed, IFNγ induced *IFITM3*

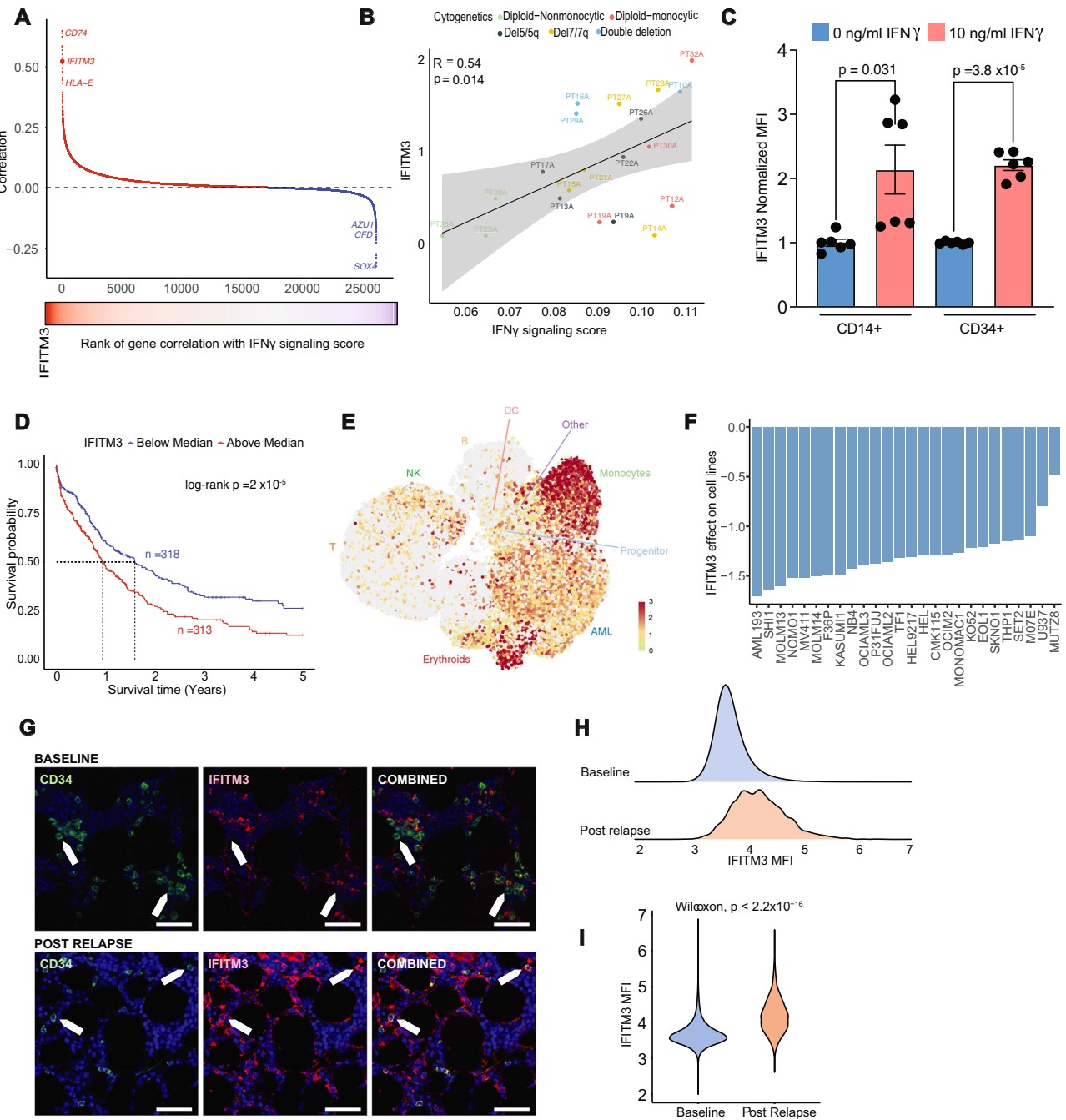

**Fig. 5 | *IFITM3* is prognostic in newly diagnosed AML patients and a potential dependency in AML cells. A** Correlation of IFNγ signaling score with gene expressions in the single cell data. Genes in red have a positive correlation and genes in blue have a negative correlation. **B** Correlation of *IFITM3* expression with IFNγ signaling score at an individual patient level. Error band represents 95% confidence interval. *T* test was used to evaluate the significance of Pearson correlation. **C** IFITM3 expression in AML blasts was assessed by flow cytometry after a 24-h stimulation with 10 ng/ml IFNγ. Data represents results from 2 patients' CD14+ blasts and 2 patients' CD34+ blasts, with each condition having 3 replicates. The error showed standard error of mean. Two-sided *t* test was used. **D** Kaplan-Meier survival curve of the AML patients from TCGA, Beat-AML and MDACC by median *IFITM3* expression in the bulk RNA profiling data. **E** UMAP projection of all patients' cells scored by IFITM3 expression in cells. **F** Change in cell line fitness following genetic deletion of *IFITM3* in the DepMap CRISPR knockout screen data, showing results in the 26 AML cell lines tested. **G** Representative multiplex IF panel. Baseline AML sample with CD34 positive blasts show low amounts of IFITM3(Top). Post relapse AML sample with CD34 positive blasts show high amounts of IFITM3 (Bottom). Green is CD34, red is IFITM3, blue is DAPI. Scale bar 50 μm. The experiment was repeated 2 times. **H** Density plot of IFITM3 fluorescence intensity on AML blasts at diagnosis and post relapse. **I** Violin plot of fluorescence intensity on AML blasts at diagnosis and post relapse. Post relapse AML sample show significantly higher amounts of IFITM3. Two-sided Wilcoxon test was used. Source data are provided as a Source Data file.

expression in both CD14+ and CD34+ leukemic blasts by 2.1 ($p = 0.0312$) and 2.2 ($p < 0.0001$) fold increase, compared to unstimulated CD14+ and CD34+ cells, respectively (Fig. 5C). This establishes a direct link between IFNγ and *IFITM3* expression in AML blasts. Cox proportional hazard and Kaplan-Meier models of AML patients integrated from the TCGA, BeatAML and MDACC cohorts revealed that patients with high *IFITM3* (>median) expression had significantly worse overall survival (Fig. 5D; Supplementary Fig. 6A–D). High *IFITM3* expression remained

significantly associated with worse overall survival with adjustments for age, blast percentage, and cytogenetic risk in a multivariable cox model (Supplementary Fig. 6E). Of note, *IFITM3* was almost exclusively expressed in AML cells, with the highest expression in monocytes and limited expression in T cells and other microenvironmental cells (Fig. 5E). This suggests that *IFITM3* is a downstream target of IFNγ, and increased expression adds valuable prognostic information to stratify newly diagnosed AML patient outcomes.

We next sought to validate the potential impact of *IFITM3* loss in AML cell lines. From the DepMap Public Dataset, which consists of 1,086 cell lines and 17,386 genes, we focused our analysis on 26 AML cell lines and 188 genes that overlapped with the IFNγ response signature in the HALLMARK gene set[52]. Unsupervised clustering applied to the gene effect identified a cluster of 7 genes (*WARS1, IFITM3, NAMPT, PSMA2, NUP93, PSMB2,* and *PSMA3*) that consistently decreased the fitness of AML cells after knockout (Supplementary Fig. 6F, G). In particular, the knockout of *IFITM3* caused the cell death in all AML cell lines tested (Fig. 5F). The significance of this change was verified by a two-sided *t* test comparing the fitness score of each gene to those of the remaining 187 genes. To further evaluate its effect on AML cells, we used multiplex immunofluorescence to measure the fluorescence intensity of *IFITM3* on blasts in baseline and post relapse samples. A total of 15 TMA cores were stained for CD34, CD56, CD45, CD4, CD14, and IFITM3 based on markers determined by flow to identify blasts (Fig. 5G; Supplementary Table 5). The fluorescence intensity in post relapse samples was significantly higher than in baseline samples ($p < 2.2 \times 10^{-16}$) (Fig. 5H, I), supporting the association of *IFITM3* with disease resistance. These data suggest *IFITM3* as a potential dependency in AML cells and could be associated with therapeutic resistance. However, more experimental data will be further needed to validate the role of IFITM3 as a mediator for drug resistance.

### IFNγ signaling confers venetoclax resistance

We next assessed the correlation of IFNγ with drug response. We leveraged the BEAT-AML ex vivo drug sensitivity data which provided detailed matched clinical, genomic, and transcriptomic analyses[3] (Supplementary Fig. 7A). Given that AML patients with monocytic differentiation and del7/7q are reported to have resistance to venetoclax-based therapy[10,53], we evaluated the correlation of IFNγ signaling score with drug sensitivity. We found a strong positive correlation between IFNγ signaling score and venetoclax resistance, indicating that IFNγ signaling confers venetoclax resistance (Fig. 6A). This finding was validated in an independent cohort[54] of ex vivo drug screening in AML, where patients with high *IFITM3* expression or IFNγ signaling score were less sensitive to venetoclax-based therapy (i.e. more resistant to venetoclax) (Fig. 6B). To further validate the role of IFNγ signaling in AML cell survival and drug resistance, we isolated leukemic blasts from primary patient samples ($n = 3$ patients) and cultured them in the absence or presence of IFN-γ, and with increasing concentrations of venetoclax then assessed the AML cell viability. IFNγ induced proliferation and higher resistance to venetoclax, confirming that IFNγ promotes survival and venetoclax resistance of AML blasts (Fig. 6C–E).

Despite its correlation with venetoclax resistance, the IFNγ signaling score did not predict survival outcomes in our bulk cohort (Supplementary Fig. 7B), suggesting that the whole gene set may contain genes that are less sensitive for predicting outcomes. Therefore, to improve the prognostic sensitivity of the IFNγ signaling score, we defined a light weighted IFNγ signature using the least absolute shrinkage and selection operator (LASSO) model[55] (Supplementary Fig. 7C). After LASSO regression, 47 genes related to survival were retained, forming a parsimonious IFNγ signature (Supplementary Table 6). We then scored the bulk AML patients data with the new parsimonious IFNγ score and revealed a tight positive correlation with

HLA class 1 and 2 scores (Supplementary Fig. 7D, E). Importantly, this parsimonious IFNγ score was able to predict patient outcomes in our bulk cohort, whereby a higher score predicted worse survival (Fig. 6F, Supplementary Fig. 7F). This finding of worse overall survival was further validated in an independent dataset[54] (Supplementary Fig. 7G). The parsimonious IFNγ score remained a significant and strong predictor of survival in a multivariate model when we accounted for age, blast percentage, and cytogenetics, with the highest hazard ratio of all these predictors (Fig. 6G). Ultimately, these results suggest that the IFNγ pathway is associated with resistance to venetoclax-based therapy and can predict patient outcomes independently of known risk factors, making it a promising target for therapeutic intervention.

## Discussion

Dissecting inflammatory states within the immune microenvironment of AML is likely to uncover mediators of therapeutic resistance and disease progression, and to aid in the development of novel immunotherapeutics. It has been demonstrated that the proliferation of aberrant myeloid cells leads to overproduction of pro-inflammatory cytokines, many of which have been established to play immunomodulatory roles that drive leukemic progression[56–59]. However, analyses of inflammation in the BM niche have been largely focused on immune cells, and the characterization of inflammatory states in AML cells remains understudied. Here we presented a comprehensive characterization of inflammation in AML using independent bulk and scRNA profiling studies of newly diagnosed AML bone marrows. With this approach we were able to uncover conserved IFNγ signaling which was more active in monocytic and del7/7q AML leading to a distinct immune microenvironment. IFNγ is one of the main mediators of inflammation in cancer and development of an IFNγ signaling score led to the identification of *IFITM3* as both a prognostic biomarker and a direct target of IFNγ with a potential dependency in AML cells lines, i.e., its deletion led to loss of AML cell fitness. Importantly, IFNγ signaling score correlated strongly with venetoclax resistance. Finally, a parsimonious IFNγ gene signature added crucial prognostic information to newly diagnosed AML patients.

IFNγ has long been recognized as a pivotal cytokine that activates cellular immunity and stimulates antitumor immune responses. However emerging studies have also indicated that it plays a role in promoting cancer immune evasion and resistance to chemotherapeutic agents[25–28,60,61]. We identified IFNγ signaling as a prominent feature in AML using bulk RNA sequencing. With scRNAseq, we were able to further disentangle the complex interplay between the AML cells and the immune microenvironment. This analysis uncovered high IFNγ pathway signaling within AML patients with monocytic differentiation and del7/7q. Our data suggested that CD8 T cells and NK cells are the source of IFNγ in the microenvironment; however, true cytokine production data are difficult to discern using scRNA-seq[62]. The antigen presentation machinery, a known downstream target of IFNγ signaling, was correlated with IFNγ signaling, but interestingly, samples with high IFNγ signaling also had high expression of HLA-E, in particular samples with monocytic differentiation. Additionally, our analysis revealed that leukemic cells expressing HLA-E were in closer proximity to T cells compared to HLA-E negative cells. This observation supports the hypothesis that tumor immune evasion mechanisms may be at play, wherein HLA-E expressing cells potentially interact with and modulate the immune response, thereby promoting their survival within the tumor microenvironment. Further research into this relationship could shed light on novel therapeutic strategies to enhance anti-tumor immune responses. The HLA-E-NKG2A axis is an emerging immune checkpoint that restrains anti-tumor immune responses and high HLA-E expression correlates with a T cell inflamed microenvironment[46,63]. Pancreatic cancer metastases are in fact mediated via cancer cell upregulation of HLA-E as a mean to evade immune recognition during transit to metastatic sites[64]. Additionally, CD74 which provides the

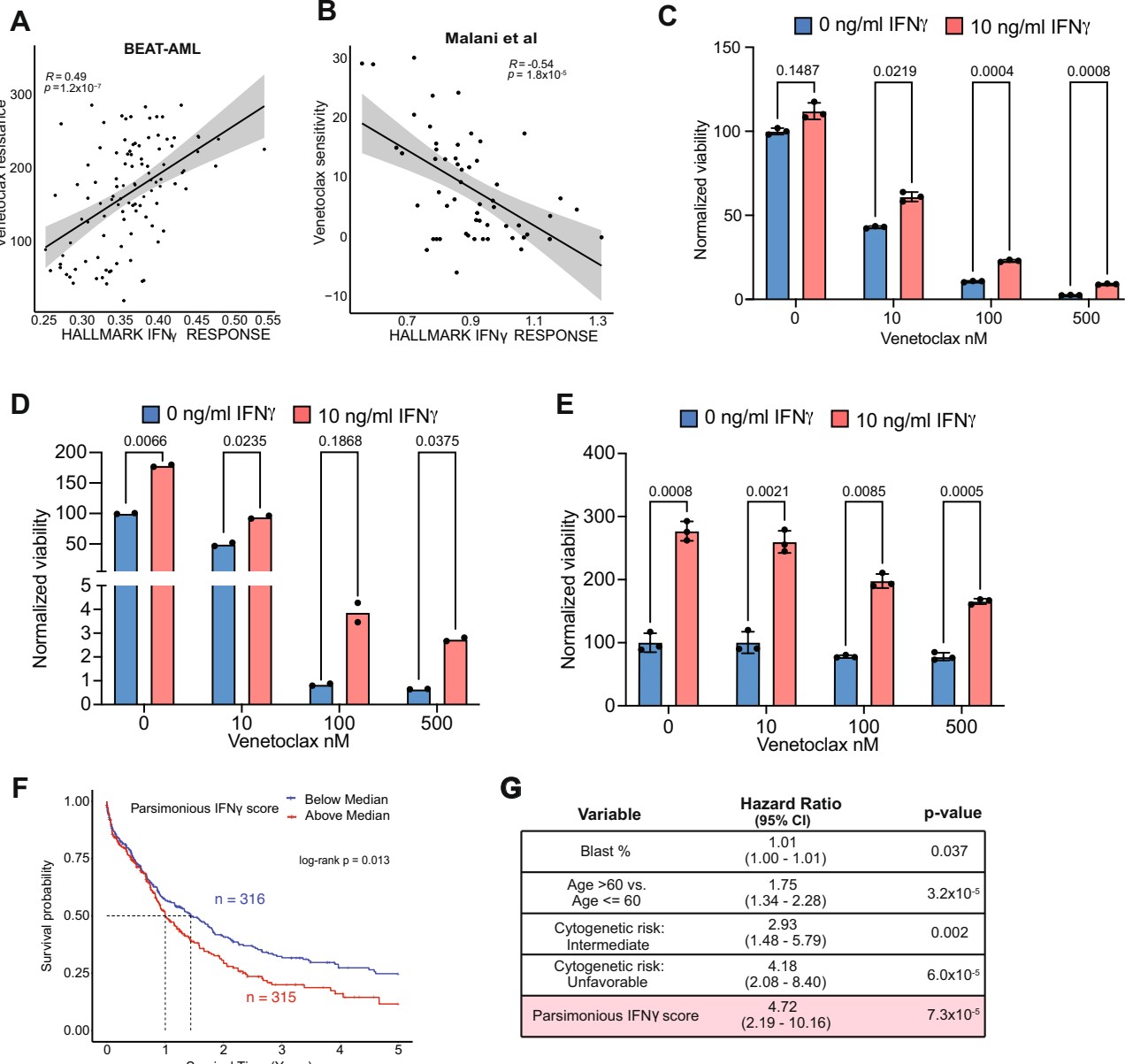

**Fig. 6 | IFNγ signaling in AML is associated with venetoclax resistance and parsimonious IFNγ score is associated with worse overall survival.**
**A** Correlation of IFNγ signaling score and venetoclax resistance in BEAT-AML data. Error band represents 95% confidence interval. *T* test was used to evaluate the significance of Pearson correlation. **B** Correlation of IFNγ signaling score and venetoclax resistance in Malani et al. data. Error band represents 95% confidence interval. *T* test was used to evaluate the significance of Pearson correlation. **C–E** Viability assessment of AML blasts after stimulation with 10 ng/ml IFNγ and incubation with venetoclax using the CellTiter-Glo luminescent cell viability assay.

Data represents results from 3 AML patients' blasts, with each condition having 3 replicates, except for (**D**), which has two replicates. For (**C**, **E**), data are presented as mean values ± SEM. Two-sided *t* test was used. **F** Kaplan-Meier survival curves of the AML patients from TCGA, Beat-AML, and MDACC by parsimonious IFNγ score in bulk RNA profiling cohort. **G** Multivariable adjusted Cox regression model for overall survival among the patients in the combined bulk RNA profiling cohort adjusting for parsimonious IFNγ score, age, blast percentage, and cytogenetics. Wald test was used to measure the significance of factors. Source data are provided as a Source Data file.

peptides for class II-associated invariant chain peptide (CLIP) was also highly correlated with IFNγ signaling. CLIP-positive AML blasts have been shown to evade CD4 T cell killing by blocking the presentation of endogenous leukemia-associated antigens and the induction of a leukemia-specific T-cell response[65,66]. Thus, our findings are consistent with emerging data that suggest that T cell- and NK cell-derived IFNγ helps to create an immunosuppressive microenvironment through HLA-E and CD74 upregulation on target cells[67–69]. In a recent study highlighting the possible detrimental effects of IFNγ in hematologic disorders, it was shown that IFNγ is not essential for CAR-T therapy. Instead, inhibiting IFNγ reduces checkpoint blockade expression,

enhances CAR-T cell proliferation and reduces associated toxicities, without mitigating CAR T- cell antitumor efficacy. Given that myeloid cells are a major target of IFNγ[70] and IFNγ promotes myeloid differentiation of HSPC progenitors[71], it is possible that myeloid leukemia, in particular, evolves to co-opt IFNγ signaling, ultimately hijacking *HLA-E* for immune evasion and promoting blast growth in the bone marrow niche.

Further analyses identified a strong correlation between *IFITM3* expression and IFNγ signaling. Interestingly, *IFITM3* expression was mainly limited to AML cells and was not seen in other bone marrow cell populations. *IFITM3* expression was also independently associated

with overall survival, as AML patients with higher expression of *IFITM3* had worse survival. The role of *IFITM3* in AML is not well described; however, a recent report confirmed our finding that high expression correlated with worse survival in AML[72]. Elevated expression of *IFITM3* has been noted in B-cell acute lymphoblastic leukemia (B-ALL) and B-cell lymphomas, and B cell malignancies with high *IFITM3* expression have poorer prognoses[51]. In models of B-ALL, *IFITM3* amplified oncogenic PI3K signaling to contribute to B-cell oncogenesis[51]. The higher expression of IFITM3 at time of relapse and the genetic loss-of-function data suggested a potential dependency of AML cells on *IFITM3*, wherein loss of *IFITM3* caused decreased fitness in all AML cell lines tested. Lastly, we were able to link IFNγ signaling with venetoclax resistance in primary patient samples. This fits with emerging data suggesting that monocytic differentiation is a major venetoclax resistance pathway[10,11]. However, further experimental validation that directly evaluates the impact of IFITM3 loss in an immunocompetent mouse model can shed further light on the direct impact of IFNγ-IFITM3 axis mediated from the tumor immune microenvironment on therapeutic resistance in AML. Though the exact mechanisms by which monocytic differentiation leads to venetoclax resistance are yet to be elucidated, our data suggest that IFNγ signaling may play a pivotal role and is a potential therapeutic target in venetoclax resistance in AML.

While our study offers important insights, it is subject to certain limitations. Notably, while we demonstrated that IFNγ secretion by T/NK cells can be recapitulated in vitro, the complexity of immune interactions within the microenvironment may extend beyond our models. The potential role of IFITM3 in AML in relation to venetoclax requires direct evidence, and the prominence of IFNγ in our findings does not exclude the possible influence of other inflammatory pathways. These factors highlight the need to further model and validate our conclusions.

In summary, through integration of bulk and scRNA profiling data from newly diagnosed AML patients, we observed high IFNγ signaling in monocytic and del7/7q AML samples which held prognostic value. These findings offer important insights into AML biology and may lead to the identification of potential novel therapeutic targets.

## Methods

The research fully complies with principles of the Declaration of Helsinki. A written informed consent for all uses of human material was approved by the MD Anderson's institutional review board.

### Bulk RNA-seq analyses

RNA-seq data and corresponding clinical data from TCGA, BeatAML and MDACC were downloaded and integrated. Batch effects were corrected using ComBat-seq[73]. Analysis was limited to adult patients with newly diagnosed AML that did not fall into FAB classification M3/M6/M7. Cytogenetic groups were determined from the clinical metadata of patients: the diploid-monocytic group include diploid patients with FAB classification M4/M5; diploid-nonmonocytic included diploid patients with FAB M0, M1, M2; and diploid-NOS referred to diploid patients with cytogenetics not otherwise specified. Gene counts were transformed into transcripts per million (TPM) for quantification of ssGSEA scores using the GSVA software package[74]. Curated gene sets (HALLMARK, KEGG, GO) for pathways were obtained from MSigDB (http://software.broadinstitute.org/gsea/msigdb/index.jsp). T cell dysfunction signatures were obtained from published literatures: T cell dysfunction score[75], T cell exhaustion score[76], and T cell senescence score[77].

### Immune-cell deconvolution

Immune infiltration levels of bulk data were predicted by CIBERSORTx[35] using its website server (https://cibersortx.stanford.edu). The TPM-normalized matrix was used as the input mixture file, and the signature matrix for 22 immune cells LM22 was used as the signature gene file to estimate the cellular components. Other settings include 100 permutations and disabled quantile normalization.

### Evaluation of cytokines in newly diagnosed AML patients

The serum levels of cytokines were quantified using MILLIPLEX Human Cytokine/Chemokine Panel 1 (MilliporeSigma, St. Charles, MO). The Luminex 3D instruments were used for the assay. To avoid the prozone effect, any samples with an initial measurement exceeding 5000 pg/ml were retested. These samples were re-evaluated after a 1:10 dilution was applied, and the results were back-calculated with the dilution factor.

### Human participants

Patients 18 years of age or older with a new diagnosis of AML seen at The University of Texas MD Anderson Cancer Center were included in this study. Written informed consent from all participants was obtained, and the study protocol was approved by MD Anderson's institutional Review Board. The study was conducted in accordance with the principles of the Declaration of Helsinki.

### Sample collection and preparation

Bone marrow biopsies were routinely collected before treatment initiation. Samples were freshly frozen with freezing medium containing 20% fetal calf serum (FCS) and 10% dimethyl sulfoxide (DMSO) in Dulbecco's modified Eagle medium (DMEM) and stored in liquid nitrogen. All frozen BM samples were retrieved immediately before sample processing. To maximize the recovery of the cellular viability, samples were processed in batches according to a thawing protocol. Briefly, cells were gently thawed in a water bath at 37 C until partially thawed and then immediately placed on ice. Next, cells were gently transferred to 10 mL of RPMI1640 supplemented with 20% fetal bovine serum [FBS] and centrifuged (453 *g* for 5 min). After removal of the supernatant, the cell pellet was carefully resuspended in 10 mL of thawing medium RPMI1640 + 20% FBS supplemented with 1 mg of heparin [stock 2 mg/ml, Cat# 9041-08-1; Sigma Aldrich], 20 μL DNase [stock 1 μg/ml, Cat# 89835, Thermo Fischer Scientific] and 200 μL MgSO$_4$ [stock 200 mM, Sigma Aldrich], followed by incubation at 37 °C for 15 min. After incubation, cells were centrifuged and gently washed twice in 2 mL of 0.04% bovine serum albumin in phosphate-buffered saline (PBS). Cells were then passed through a 35μm strainer to remove cell clumps. Finally, cells were stained with 10 μL of 0.4% Trypan blue and quantified and assessed for viability using the standard hemocytometer and light microscope. Cells were eventually centrifuged again and re-suspended in the appropriate amount of PBS to adjust for the desired cell density.

### Library preparation and scRNA-seq

The 5' gene expression libraries were prepared using a 10x Genomics Chromium Controller instrument and Chromium Single-Cell 5' V5.1 reagent kits (10x Genomics). Briefly, cells were concentrated to 1000 cells/μL and loaded into each channel to generate single-cell gel bead-in-emulsions, resulting in mRNA barcoding of an expected 10,000 single cells for each sample. After the reverse transcription step, gel bead-in-emulsions were broken, and single-strand cDNA was cleaned with DynaBeads. The amplified, barcoded cDNA was fragmented, A-tailed, ligated with adaptors, and amplified by index polymerase chain reaction. A High-Sensitivity D5000 DNA Screen Tape analysis (Agilent Technologies) and the Qubit dsDNA HS Assay Kit (Thermo Fisher Scientific) were used to assess cDNA and the constructed libraries. Sequencing was conducted on an Illumina NovaSeq6000 sequencer with 2 × 100 bp paired reads to target sequencing depth of 50,000 read pairs per cell.

### scRNA-seq data processing

Raw scRNA-seq data were demultiplexed and aligned and read count matrix was generated using the Cell Ranger Single Cell Software Suite

provided by 10x Genomics. The Seurat v4 (version 4.2.1) R package[78] was used to analyze the scRNA-seq data. Detailed quality metrics were evaluated. Genes detected in fewer than 3 cells and cells with fewer than 250 genes, or 500 transcripts were excluded from subsequent analysis. Cells with more than 10% of reads mapping to mitochondrial genes were removed as well. Possible doublets were identified and removed by (1) the DoubletFinder package[79] (2) identifying cells expressing markers of distinct cell types, and (3) identifying cells exhibiting aberrantly high gene counts. Cells from all samples were then merged and normalized. Batch effects were removed using the Harmony package[80] before clustering analysis. The filtered count matrix was normalized with the *NormalizeData* function and was used to identify the most variable features using a variance stabilizing transformation. Cell-cycle scores ("S.Score" and "G2M.Score") and the cell cycle phase were calculated and assigned to each cell using the *CellCycleScoring* function. Cell cycle effects were regressed out using the *ScaleData* function. Principal component analysis was performed on variable features and an elbow plot was generated to assess the optimal number of principal components. Uniform Manifold Approximation and Projection (UMAP) layouts and nearest-neighbor graphs were generated using the top 20 components. Different resolutions for graph-based clustering were examined to determine the best number of clusters.

### Inference of copy number variations

For patients with clinically abnormal karyotypes, the inferCNV tool was used to infer the large-scale copy number variations (CNVs)[81]. Normal monocytes from 3 healthy donors were used as controls. InferCNV was run on each patient with nondiploid cytogenetics separately using default settings except cutoff=0.1 and denoise=TRUE.

### Determination of cell type and AML cell states

To define the major cell types, the *FindAllMarkers* function from Seurat was used to identify differentially expressed genes (DEGs) for each cluster. The top 50 DEGs for each cluster and a set of canonical marker genes were carefully compared to assign cell types. Flow cytometry and inferred CNVs were used to further confirm the identity of malignant cells in patients with nondiploid cytogenetics. Of note, for the 20 patients included in this study, CD8 T cells ($n = 10,777$ cells) were recently analyzed as part of another study[75]. AML cells were subset from each patient and integrated. The cellular states of AML cells were determined by the cell type classifier described by Zeng et al.[32] using their 7 predefined leukemic cell populations spanning the differentiation trajectory: LSPC-Quiescent, LSPC-Primed, LSPC-Cycle, GMP-like, ProMono-like, Mono-like and cDC-like.

### Regulon activity analysis

From each of the 5 cytogenetic groups in single cell data, 10% of cells were subsampled, and the raw count matrix of subsamples was used as the input for SCENIC to predict gene regulatory networks (regulons) in R[39]. Coexpression modules between transcription factors and target genes were inferred, followed by regulon selection based on enrichment of binding motifs. Regulon activities were scored using AUCell in each individual cell. The top 15 differentially activated regulons in each cytogenetic group were selected by two-sample *t* test and visualized by ComplexHeatmap[82].

### Cellular communications

To analyze cell-cell interactions among AML, CD4, CD8 and NK cells, we applied MultiNicheNet, which is an extension of the original NicheNet pipeline that better predicts niche-specific ligand-receptor (L-R) pairs[48]. DEGs between cytogenetic groups were calculated. Genes with log2 fold change >0.5 and adjusted *P* value < 0.05 were selected as gene sets of interest. All expressed genes in our sample were used as background genes. Genes were considered as expressed when they had non-zero values in at least 10 of the cells in a cell type. Sender and receiver were set on all cell types to obtain a comprehensive L-R network. The top 100 active L-R pairs across all groups and the top 50 in the diploid monocytic group were selected and visualized via circos diagrams and heatmaps using the *make_circos_group_comparison* and *make_sample_lr_prod_activity_plots* functions.

### Spectral flow cytometry

Patient peripheral blood mononuclear cell samples that had been frozen in 90% FBS with 10% dimethyl sulfoxide were obtained from the MD Anderson Leukemia Sample Bank. R20 media was made using RPMI 1640 (Corning), 20% heat inactivated FBS (Sigma Aldrich), 1% Penicillin/Streptomycin (Sigma Aldrich), 1% HEPES buffer (Corning), and 1% Glutamax (Gibco). Samples were thawed in 10 mL warm media (40% FBS + R20) and incubated with 0.5 mL of Benzonase® (EMD Millipore) to remove cell clumps, and 1–2×10⁶ cells were resuspended in Phosphate Buffer Saline (PBS; Corning) for staining. The cells were incubated with Zombie Aqua™ viability stain (1:1000; BioLegend) for 15 min at 4 °C, followed by Fc blocking with 1:300 human TruStain FcX™ (BioLegend), 2% normal human serum, and 2% normal mouse serum (15 min, 4 C). A surface staining cocktail was prepared using Brilliant Stain Buffer (BD Biosciences), and cells were incubated for 15 min at 4 C in the dark. Fixation was done using 4% methanol-free paraformaldehyde (ThermoFisher; 15 min, 4 C), followed by an incubation with Perm/Wash Buffer (BD Biosciences; 15 min, 4 C) and intracellular staining cocktail (20 min, 4 C). Cells were resuspended in 300 mL PBS for spectral flow cytometry using the 5-laser Cytek Aurora system. Data analysis was performed in FlowJo™ 10.8.2. Fluorescently labeled anti-human antibodies against CD3 (HIT3a), CD33 (WM53), CX3CR1 (2A9-1), CD14 (M5E2), CD64 (10.1), CD74 (LN2), HLA-E (3D12), CD34 (581), HLA-DR/DP/DQ (Tü39), and IFNγ (4 S.B3) were obtained from BioLegend, and CD4 (SK3) and CD8 (RPA-T8) were obtained from BD Biosciences.

### Multiplex immunofluorescence imaging

The Lunaphore Comet multiplex IF platform was used to profile the leukemic and immune bone marrow microenvironment. A formalin-fixed, paraffin embedded tissue microarray slide containing 75 cores from 12 patients was baked, dewaxed, treated with 3% hydrogen peroxide, and rehydrated. Antigen retrieval was performed at 107 C for 15 min in an ethylenediaminetetraacetic acid-based buffer. TrueBlack lipofuscin autofluorescence quencher was applied for 1 min before the slide was washed in PBS. The slide was then loaded into the Lunaphore Comet to fit 16 cores (4 × 4) in a 9 × 9 mm square imaging window to capture two patients enriched in CD34⁺ leukemic blasts. Staining, imaging, and elution was performed cyclically by the Comet until all antibodies were captured. The resulting tiff files were imported into the Visiopharm image analysis software. Cell segmentation was performed using U-Net. Mean intensity thresholding was used to phenotype cells. Distances between cell types were calculated in Rv.4.2.1.

### AML blasts and T cells co-culture assay

Human T cells ($5 × 10^4$ per well) isolated from peripheral blood mononuclear cells (PBMCs) from a healthy donor using the EasySep Negative Human T Cell Kit (Cat# 19051, STEMCELL Technologies) were rested in R20 supplemented with 200 U/ml recombinant human IL-2 for 72 h. CD34+ blasts and CD14+ blasts were isolated from patients PBMC using the EasySep Human CD34 Positive Selection Kit II (Cat# 17856, STEMCELL Technologies) and the EasySep Human CD14 Positive Selection Kit II (Cat# 17858, STEMCELL Technologies), respectively. AML patients' blasts from one CD34+ patient (myeloid), one CD14+ patient (monocytic), THP1 cells, and MOLM13 cells were co-cultured ($5 × 10^4$ per well) with the healthy donor's T cells ($5 × 10^4$ per well) in R20 media in a U-bottom 96-well plate. Additionally, wells containing blasts alone, T cells alone, or a 1:1 ratio of T cells with

Dynabeads Human T-Activator CD3/CD28 [Cat# 1161D, Thermo Fisher] were included for comparison. After 72 and 120 h of culture, the culture media were collected, and IFN-γ levels were assessed by ELISA using the LEGEND MAX™ Human IFN-γ ELISA Kit [Cat# 430107, BioLegend].

### AML blasts IFN-γ stimulation assay

To assess IFITM3 levels, CD34+ blasts and CD14+ blasts cultured in R20 media ($5 \times 10^5$ per well in a 24-well plate) were stimulated with 10 ng/ml of human recombinant IFN-γ [Cat# 300-02-20UG, PeproTech]. Unstimulated (PBS) blasts were included for comparison. After 24 h of culture, the cells were collected, washed twice with PBS, and intracellularly stained for IFITM3 as described in the Spectral Flow Cytometry protocol with minor modifications. Instead of the intracellular staining cocktail, cells were incubated with a 1:250 IFITM3 monoclonal antibody (1 h, room temperature) [Cat# D8E8G, Cell Signaling Technology], followed by washing and incubation with a 1:1000 Alexa Fluor™ Plus 488 conjugated secondary antibody (1 h, room temperature) [Cat # A32731, Invitrogen]. The cells were washed and resuspended in 300 μL of PBS for flow cytometry using the Beckman Coulter Gallios Flow Cytometer. Data analysis was performed in FlowJo™ 10.8.2."

### Treatment of AML blasts with Venetoclax

CD34+ blasts cultured in R20 media were stimulated with 10 ng/ml of human recombinant IFN-γ. Unstimulated (PBS) blasts were included for comparison. After 12 h of culture, blasts with or without IFN-γ stimulation were counted and plated in a U-bottom 96-well plate ($5 \times 10^4$ per well), with or without 10 ng/ml of human recombinant IFN-γ, along with titrated concentrations of venetoclax [Cat# S8048, Selleck Chemicals]. After 24 h of incubation, cell viability was assessed using the CellTiter-Glo luminescent cell viability assay [Cat# G7572, Promega] following the manufacturer's protocol. IC50 values were calculated using GraphPad Prism version 10.1.0.

### Evaluation of gene effects on AML cell lines by CRISPR

The CRISPR data for cell lines were downloaded from DepMap Public 22Q4 (https://depmap.org/portal/). Gene effects were analyzed by DepMap using Chrono, which infers gene fitness based on cell proliferation. DepMap metadata was used to separate the AML cell lines. A total of 26 AML cell lines were selected to evaluate the 188 IFNγ response pathway (HALLMARK) genes that overlapped with the CRISPR data. The gene effect value was used to create the heatmap using the pheatmap package in R. Positive values indicate that knocking out the gene leads to the growth of AML cell lines, while negative values represent cell death in AML cell lines. The gene dependency scores were evaluated by a two-tailed $t$ test using the Empirical Bayes Statistics for Differential Expression implemented in the limma R package[83] and presented by volcano plot.

### LASSO regression for feature selection

A LASSO regression model was applied to gene expression data for AML patients to select genes correlated with survival. The model was implemented in the R package glmnet[84] and trained using tenfold cross-validation to select the best penalty parameter. Genes with a non-zero coefficient were retained.

### Statistical analysis and reproducibility

All statistical analyses were performed using R v.4.2.1 and GraphPad Prism10. Pearson correlation tests were applied to assess the relationship between pathway enrichment scores and gene expressions. Kruskal-Wallis and Wilcoxon rank sum tests were used to compare IFNγ signaling in different cytogenetic groups. Log-rank test was used in comparing survival curves. All statistical significance testing in this study was two-sided, and results were considered statistically significant at $P$ values or FDR $q$-values (for multiple testing) was <0.05. Default "$P < 2.2 \times 10^{-16}$" in R v4.2.1 was used if the $P$ value was too small to illustrate.

Sample sizes were determined with the intent to capture a representative cohort of AML patients, encompassing the principal molecular subgroups identified in current research. This decision was also informed by practical considerations including budgetary constraints and the availability of samples within our databank. Due to the nature of our study, which necessitated the correlation of molecular findings with clinical outcomes, randomization and blinding were not employed.

### Reporting summary

Further information on research design is available in the Nature Portfolio Reporting Summary linked to this article.

## Data availability

The single cell sequencing data generated in this study have been deposited in the GEO database with the accession number (GSE239721). The metadata and raw counts of bulk RNA sequencing data were downloaded from https://gdc.cancer.gov/about-data/publications/pancanatlas (TCGA), GEO database GSE165656 (MDACC) and https://www.nature.com/articles/s41586-018-0623-z#Sec38 (BEAT-AML). The public drug screening data were download from https://zenodo.org/records/7274740 and https://biodev.github.io/BeatAML2/. Source data are provided with this paper.

## Code availability

All codes can be accessed from https://github.com/abbaslab/2023_IFNG_Inflammation[85].

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

## Acknowledgements

H.A.A. is supported by Elsa U. Pardee Foundation, American Society of Clinical Oncology Young Investigator Award, Leukemia Research Foundation and Physician Scientist Award. P.K.R. is supported on a postdoctoral fellowship from the Lymphoma Research Foundation. M.R.G is supported by a Scholar award from the Leukemia and Lymphoma Society. Sequencing was performed by MD Anderson's Advanced Technology Genomics Core (ATGC), which is supported in part by NIH CCSG grant P30 CA016672 (ATGC), and NIH S10 grant 1S10OD024977-01. Flow cytometry was performed in the Flow Cytometry & Cellular Imaging Core Facility, which is supported in part by the National Institutes of Health through M. D. Anderson's Cancer Center Support Grant P30 CA016672. We would like to acknowledge the Editing Services, Research Medical Library at The University of Texas at MD Anderson Cancer Center for the critical review and comments of the manuscript.

## Author contributions

H.A.A. conceived the study, supervised all aspects of the work, co-wrote and reviewed the manuscript. B.W, P.K.R and H.A.A. wrote the manuscript. B.W. led the computational analyses. P.K.R co-analyzed the data. M.Y.Y and Z.W conducted the experiments. Q.D. and M.R.G. led the library preparation. F.Z.J. and G.T. performed clinical and hematopathologic annotations. J.M. and A.I.G contributed to patient sample collection. N.B., P.H.G, D.A.A., S.M.P. N.R.V. and K.S. contributed to data analysis. P.N.D., P.B., C.L., E.D., J.B., I.V and C.N. contributed to data analysis, data visualization, experimental design and/or writing the manuscript. B.C., S.E., M.B.G., S.G., D.H., A.F., M.K. and N.D. contributed conceptually to data analysis and design. All authors edited the manuscript.

## Competing interests

M.R.G reports research funding from Sanofi, Kite/Gilead, Abbvie and Allogene; consulting for Abbvie, Allogene and Bristol Myers Squibb; honoraria from BMS, Daiichi Sankyo and DAVA Oncology; and stock ownership of KDAc Therapeutics. I.V. received research funding from Avilect Biosciences/Aviceda Therapeutics. H.A.A. reports research funding from Genentech and Enzyme-By-Design, consultancy fees from Molecular Partners, and inkind support from Illumina.

## Additional information

**Peer review information** : *Nature Communications* thanks Jiyang Yu, John Pimanda and the other, anonymous, reviewer(s) for their contribution to the peer review of this work. A peer review file is available.

[1]Department of Leukemia, Division of Cancer Medicine, The University of Texas MD Anderson Cancer Center, Houston, TX, USA. [2]Department of Hematopathology, Division of Pathology & Laboratory Medicine, The University of Texas MD Anderson Cancer Center, Houston, TX, USA. [3]School of Biomedical Informatics, The University of Texas Health Science Center, Houston, TX, USA. [4]Department of Biology and Biochemistry, University of Houston, Houston, TX, USA. [5]School of Basic Medical Sciences, Harbin Medical University, Harbin, Heilongjiang, China. [6]Department of Lymphoma & Myeloma, Division of Cancer Medicine, The University of Texas MD Anderson Cancer Center, Houston, TX, USA. [7]Department of Melanoma Medical Oncology, Division of Cancer Medicine, The University of Texas MD Anderson Cancer Center, Houston, TX, USA. [8]Department of Pediatrics, Division of Pediatrics, The University of Texas MD Anderson Cancer Center, Houston, TX, USA. [9]Department of Genomic Medicine, Division of Cancer Medicine, The University of Texas MD Anderson Cancer Center, Houston, TX, USA. [10]Department of Oncology, Albert Einstein College of Medicine, Bronx, NY, USA. [11]These authors contributed equally: Bofei Wang, Patrick K. Reville. ✉e-mail: habbas@mdanderson.org

