## [Peer Review File · Nature Communications]

REVIEWER COMMENTS

Reviewer #1, expertise in scRNAseq, bioinformatics, immunology and immuno-oncology (Remarks to the Author):

Inflammation has been implicated to have a dichotomous relationship with AML initiation and progression where some studies demonstrate necrotic factors and inflammatory cytokines like IL-1 β , TNF α and IL-6 increasing AML aggressiveness and IL-10 slowing AML progression. In Wang et al, the authors describe previous work that suggests INF γ promotes cancer growth and promotes T cell exhaustion and senescence. Thus, the authors looked at the INF γ and its role in AML. Subsequently, from TCGA, BeatAML and MDACC datasets, they found higher INF γ signaling score in FAB M4/M5 AML subtype. This signaling score wasn't seen in 17 healthy adult BM samples. Additionally, they found strong correlation between INF γ signaling score and T cell dysfunction (fig supp 1d-f). Then, they carried out deconvolution from bulk RNA-seq to identify cells correlated with INF γ signaling and identified monocytes to be most correlated (R= 0.4). Then they carried out scRNA-seq from 20 BM aspirates (mean age 73, 20% female and 65% de novo). ~100,000 cells passed QC and they found AML cells (as identified by copy number variation) from diploid monocytic AML had the highest INF γ signaling score relative to T cells (AML/T Cell = 0.9). The authors also note that ref 32(Zeng et al) reported that monocytic-like cells in AML express high levels of INF γ , consistent with their findings. To understand the regulatory mechanism of INF γ induction in the monocytic cells, the authors employed SCENIC to infer transcriptional factors from their scRNA-seq data and found IRF1, IRF5 and IRF8 had elevated levels, especially IRF8 was prominent in diploid AML subtypes with monocytic differentiation. They also found downstream targets of INF γ , HLA-E was upregulated in AML blast cells. They also found CD8 and NK cells the primary source of INF γ expression (supp fig 5A), and T cells have higher BM infiltration in Diploid monocyte subsets versus non-monocytic subsets (supp fig 5B). Subsequent cellchat analysis showed INF γ signaling from Cd8 T cells and NK cells communicating with monocytes from diploid monocyte subsets (fig 4b). They investigated the consequences of INF γ signaling by looking at genes that are correlated with INF γ signaling and found IFITM3 to be highly correlated in addition to B-ALL and several other cancers. Cox survival analysis also showed that patients with high IFITM3 expression have poor survival (fig5C). Then, they investigated the effects of IFITM3 gene KO by analyzing the DepMap public dataset (CRISPR KO dataset from 26 AML cell lines) and found its KO results in the decrease of fitness in some cell lines (fig5E). The authors leveraged many openly available data sets and did good work to establish answerable questions. The visuals used are also great as they are reader friendly.

Overall, I think the work is exciting but needs substantial more work before its ready for publication in Nat. Comm.

Major:

1. One of the major shortcomings of this manuscript is the lack of experimental validation of findings made from computational analysis.

2. One of the major unanswered questions from their work is the cause of increased INF γ expression by the immune microenvironment. Do the monocytic cells stimulate the CD8 and NK cells to express INF γ or there are extra-BM factors that stimulates the inflammation?
3. Experimental validation of INF γ stimulating IFITM3 expression in AML monocytic blast is necessary as well, since most of the evidence is correlation and not causal.
4. It is also typically easy to quantify INF γ in blood or BM using Immunohistochemistry/ ELISA assays. Since the authors have BM frozen from the newly diagnosed patients, I believe they should show INF γ overproduction should be demonstrated directly.
5. Another question the authors did not address is the difference the signaling has on AML monocytes and non-AML monocytes. This also needs experimental investigation in addition to showing IFITM3 is a direct signaling target of INF γ in AML- monocytes.

Minor:

1. Forgot to mention supplemental table1.
2. In content of Figure 1 E, please mention why sorting CD34+cell.
3. Around Figure3C please mention 4 distinct subtypes first, then give information that monocyte-like state has highest IFN-g signaling activity. It was hard to understand at first. (clarify)
4. In 288, We validated the IFN-g signaling activity in these patients, and it was highest in the mature state while lowest in GMP states. -> what is the mature state? (clarify)
5. Figure 3D. Show overall regulon activity of IRFs in AML (not only IRF1,5 but all or sum), and highlight IRF8.
6. Figure 3E. Show results for all 19 core transcriptional regulators.
7. In 269, the last sentence in the paragraph. Can't find the data matches to the sentence. Highest proportion of NK cell? Does percentage of NKG2A matter? Diploid non-monocytic express NKG2A the least.

Reviewer #2, expertise in AML genetics (Remarks to the Author):

Wang et al leverage public AML datasets and generate scRNA sequencing data on 20 bone marrow aspirates to perform a focused study on IFN γ signaling in AML cells and associations with AML cell, immune cell characteristics, survival and venetoclax resistance. The essence of their findings is that high IFN γ signalling was associated with monocytoid differentiation of AML cells, which also displayed up/down stream gene expression correlations that fit. Using T cell-AML cell gene expression data the authors draw associations between CD8 T cells/NK cells as producers of IFN γ and AML cells- particularly monocytic cells as receivers of these signals and perform supplementary analyses to show expression levels of select response genes with survival/VEN resistance and generate a parsimonious IFN γ response score that correlates with VEN resistance.

Overall, a nice analysis of a focused question utilizing both public and self-generated single cell data and a suite of analytical tools. Although the overall findings of monocytoid differentiation correlating with IFN γ signaling is unsurprising and the association of monocytoid LSCs as drivers of VEN + azacitidine resistance (not VEN alone and unclear how much of this is due to VEN resistance as opposed to azacitidine refractoriness due to high cytidine deaminase activity in monocytic cells) continue to be reported by Jordan and colleagues, this is solid confirmation from an independent angle.

Comments:

1. Type 1 and Type 2 INF signaling pathways overlap- was any analysis performed on Type response to contrast with INF γ responses?
2. Figure 4B is poorly described and presented and labels for 4C-D should be improved.
3. Could IFN γ signaling in AML cells be related to IFNGR1/2 cell surface expression and potentially an easier marker to assay than gene expression signatures?
4. Clarify labels in Figure 6 (VEN or VEN + AZA) and whether we are looking at VEN or VEN+ AZA resistance.

Reviewer #3, expertise in IFN γ and immune regulation (Remarks to the Author):

The authors present a gene expression dataset from a collection of bone marrow samples from newly diagnosed AML patients and imply inflammatory processes from the data. They conclude that an IFN-g signature negatively correlates with survival and drug sensitivity, a feature that is most prominent in subsets of disease characterized by distinct differentiated stage. Their data suggest that IFN-g in the tumor environment is derived largely from CD8 and NK cells. They also identify a potential IFN-g target gene, IFITM3, that might confer drug resistance and poor prognosis. Combining gene expression and survival data, they derive a minimal IFN-g gene signature that predicts patient survival, although the full IFN-g expression pattern did not correlate. They conclude that these data could be helpful in diagnosis of patients where IFN-g contributes to disease and could indicate that IFITM3 could be a target for therapy.

There are some issues that detract from enthusiasm for this report.

1. The data are largely correlative, with no direct experiments that demonstrate a causative role for IFN-g in disease progression or drug resistance.
2. The implication of a critical role for IFITM3 in AML is implied from public data and has been previously reported. No direct evidence was presented that IFITM3 confers AML cell survival or drug resistance, other than the previously published gene targeting data.
3. IFITM3 and the other genes in the IFN-g signature are also induced by other cytokines, with the most significant overlap being with IFN- γ . No data are presented to conclusively show that IFN-g is the culprit here rather than other inflammatory cytokines.
4. While IFITM3 expression correlated with AML, it was also highly expressed in monocytes, which are likely the most IFN-responsive blood subset. Whether it has a differential role in normal versus cancerous cells remains unclear.

5. It would seem possible to design ex vivo experiments to directly test the role of IFN-g and the expression of IFITM3 in AML cell survival and drug resistance.

6. It is unclear why CD74 expression is considered to be consistent with an immunosuppressive environment since its function should increase antigen presentation and the potential for cytotoxic T cell recognition.

Responses to Reviewer 1

Inflammation has been implicated to have a dichotomous relationship with AML initiation and progression where some studies demonstrate necrotic factors and inflammatory cytokines like IL-1 β , TNF α and IL-6 increasing AML aggressiveness and IL-10 slowing AML progression. In Wang et al, the authors describe previous work that suggests INF γ promotes cancer growth and promotes T cell exhaustion and senescence. Thus, the authors looked at the INF γ and its role in AML. Subsequently, from TCGA, BeatAML and MDACC datasets, they found higher INF γ signaling score in FAB M4/M5 AML subtype. This signaling score wasn't seen in 17 healthy adult BM samples. Additionally, they found strong correlation between INF γ signaling score and T cell dysfunction (fig supp 1d-f). Then, they carried out deconvolution from bulk RNA-seq to identify cells correlated with INF γ signaling and identified monocytes to be most correlated (R= 0.4). Then they carried out scRNA-seq from 20 BM aspirates (mean age 73, 20% female and 65% de novo). ~100,000 cells passed QC and they found AML cells (as identified by copy number variation) from diploid monocytic AML had the highest INF γ signaling score relative to T cells (AML/T Cell = 0.9). The authors also note that ref 32 (Zeng et al) reported that monocytic-like cells in AML express high levels of INF γ , consistent with their findings. To understand the regulatory mechanism of INF γ induction in the monocytic cells, the authors employed SCENIC to infer transcriptional factors from their scRNA-seq data and found IRF1, IRF5 and IRF8 had elevated levels, especially IRF8 was prominent in diploid AML subtypes with monocytic differentiation. They also found downstream targets of INF γ , HLA-E was upregulated in AML blast cells. They also found CD8 and NK cells the primary source of INF γ expression (supp fig 5A), and T cells have higher BM infiltration in Diploid monocyte subsets versus non-monocytic subsets (supp fig 5B). Subsequent cellchat analysis showed INF γ signaling from Cd8 T cells and NK cells communicating with monocytes from diploid monocyte subsets (fig 4b). They investigated the consequences of INF γ signaling by looking at genes that are correlated with INF γ signaling and found IFITM3 to be highly correlated in addition to B-ALL and several other cancers. Cox survival analysis also showed that patients with high IFITM3 expression have poor survival (fig5C). Then, they investigated the effects of IFITM3 gene KO by analyzing the DepMap public dataset (CRISPR KO dataset from 26 AML cell lines) and found its KO results in the decrease of fitness in some cell lines (fig5E). The authors leveraged many openly available data sets and did good work to establish answerable questions. The visuals used are also great as they are reader friendly.

Overall, I think the work is exciting but needs substantial more work before its ready for publication in Nat. Comm.

Response: We appreciate the summary of our work and the constructive feedback and valuable suggestions which have helped enhance the manuscript significantly. We hope that the revised analysis, additional experimental work, and the inclusion of additional patient data adequately address your concerns and meet your expectations.

Major:

1. One of the major shortcomings of this manuscript is the lack of experimental validation of findings made from computational analysis.

Response: We undertook additional experimental validation to fortify our work based on reviewers' questions which we detailed further in corresponding sections of our response:

- We evaluated the effect of $\text{INF}\gamma$ on primary AML patient samples, establishing that $\text{INF}\gamma$ promotes proliferation of primary AML cells *ex vivo*.
- We confirmed that $\text{INF}\gamma$ can promote resistance to venetoclax in *ex vivo* treatment of primary AML cells. The addition of $\text{INF}\gamma$ to *ex vivo* culture led to significantly impaired venetoclax killing, even at elevated venetoclax doses (500 nM).
- We performed $\text{INF}\gamma$ stimulation in AML cells from primary patient samples and confirmed that $\text{INF}\gamma$ significantly stimulated IFITM3 protein expression in leukemic cells.
- Using multiplex immunofluorescence, we measured the IFITM3 expression in post-relapse patient bone marrows compared to pretreatment/baseline bone marrows and confirmed it was higher at time of relapse, suggesting its role with therapeutic resistance and worse outcomes.
- We conducted co-culture assays of AML-T cells and established that direct AML-T cell interactions significantly enhance $\text{INF}\gamma$ production. It is important to note that when T cells were stimulated with CD3/CD28 Dynabeads, there was an even greater increase in $\text{INF}\gamma$ levels. This indicates that although AML-T cell interactions are sufficient for $\text{INF}\gamma$ production in the bone marrow, other factors may contribute to further amplifying T cell activation, as evidenced by the dynamic response to stimulation with Dynabeads.

In summation, our results support our hypothesis that T-cell mediated $\text{INF}\gamma$ has direct impact on primary AML cells and promotes resistance to venetoclax. Our experimental work also underscores the intricate dynamics of the AML microenvironment, emphasizing the multifaceted interplay between cellular components.

2. One of the major unanswered questions from their work is the cause of increased $\text{INF}\gamma$ expression by the immune microenvironment. Do the monocytic cells stimulate the CD8 and NK cells to express $\text{INF}\gamma$ or there are extra-BM factors that stimulates the inflammation?"

Response: Thank you for highlighting the need to elucidate the mechanisms behind the increased expression of $\text{INF}\gamma$ within the immune microenvironment of AML. To unravel the complexities of AML-immune cell interactions, we employed single-cell RNA sequencing analysis on samples from 20 AML patients. This analysis was designed to deconvolute the varied cellular components and to provide a detailed understanding of the interplay between AML cells and immune cells, using tools such as Multinichenet and CellChat. Our results, depicted in Figure 4, support signaling pathways from CD4, CD8, and NK cells that converge upon AML cells.

To specifically determine whether AML cells can directly stimulate $\text{INF}\gamma$ production, we conducted co-culture assays with T cells using primary patient samples and cell lines (MOLM13 and THP1). We measured $\text{INF}\gamma$ levels via ELISA at both day 3 and day 5 post-co-culture. Our findings demonstrated that AML cells cultured separately did not show an increase in $\text{INF}\gamma$ levels; however, co-culture with T cells resulted in a significant and reproducible rise in $\text{INF}\gamma$

concentration in the culture supernatant at both time points, confirming the role of AML-T cell interactions in stimulating $\text{INF}\gamma$ secretion. We also assessed T cell activation capacity by employing CD3/CD28 Dynabead stimulation. This revealed a considerably greater increase in $\text{INF}\gamma$ secretion from T cells post CD3/CD28 stimulation compared to the co-culture assay. This substantial increase suggests that, while direct AML-T cell interaction is indeed capable of initiating $\text{INF}\gamma$ production, other factors may contribute to further amplifying T cell activation. We have included these updated results in Figure 4F. We have also updated the results section to reflect these findings, as follows:

“To assess the capacity of AML cells to directly induce $\text{INF}\gamma$ secretion, we performed co-culture assays of AML cells with T cells in primary cells of two AML patients and two AML cell lines, MOLM13 and THP1. Our analysis revealed that in isolation, AML cells did not secrete $\text{INF}\gamma$, while T cells had a low basal level of $\text{INF}\gamma$ secretion (Figure 4F). In contrast, when AML cells were co-cultured with T cells, there was a consistent and significant elevation in the levels of $\text{INF}\gamma$ detected in the supernatant at both assessed time points relative to T cells in isolation (Figure 4F). This increase in $\text{INF}\gamma$ levels upon co-culture suggests that AML-T cell interactions is sufficient for $\text{INF}\gamma$ production. Of note, strong T cell activation with agonistic anti-CD3/CD28 stimulation was associated with a considerably greater increase in $\text{INF}\gamma$ secretion from T cells compared to the co-culture assay (Figure 4F). This substantial increase suggests that, while direct AML-T cell contact is capable of initiating $\text{INF}\gamma$ production, other factors may contribute to amplifying or dampening T cell activation.”

Figure 4F: $\text{INF}\gamma$ levels in culture media were assessed by ELISA after a 72 and 120-hour co-culture of CD14+, CD34+ AML blasts, THP1 cells or MOLM13 cells with healthy donor T cells. Control groups included blasts alone, T cells alone, and T cells co-cultured with DynaBeads T cell activator. Co-culture data is from 1 patient with CD14+ blasts, 1 patient with CD34+ blasts, THP1 cells or MOLM13 cells with each condition having 3 replicates.

3. Experimental validation of $\text{IFN}\gamma$ stimulating IFITM3 expression in AML monocytic blast is necessary as well, since most of the evidence is correlation and not causal.

Response: We agree with the reviewer that this is an important experiment to perform and can provide a direct link of $\text{IFN}\gamma$ to IFITM3. To address this critical point, we performed targeted experiments where isolated AML blasts were treated with $\text{IFN}\gamma$. We subsequently conducted flow cytometric analysis to quantify IFITM3 expression levels 24 hours post-treatment. Our results demonstrate a statistically significant increase in IFITM3 expression following $\text{IFN}\gamma$ stimulation, with an increase of 2.1-fold ($p=0.0312$) in CD14+ AML blasts and 2.2-fold ($p<0.0001$) in CD34+ AML blasts, compared to their respective unstimulated controls. This clear elevation in IFITM3 levels upon $\text{IFN}\gamma$ exposure substantiates a direct causal link between $\text{IFN}\gamma$ signaling and the upregulation of IFITM3 in AML blasts.

We have added these results in Figure 5C. The results section has been updated.

“To evaluate whether $\text{IFN}\gamma$ directly stimulates the expression of IFITM3 in AML cells, we isolated CD14+ and CD34+ AML blasts from two patients each and performed $\text{IFN}\gamma$ stimulation for 24 hours, followed by IFITM3 protein level measurement via flow cytometry. Indeed, $\text{IFN}\gamma$ induced IFITM3 expression in both CD14+ and CD34+ leukemic blasts by 2.1 ($p= 0.0312$) and 2.2 ($p < 0.0001$) fold increase, compared to unstimulated CD14+ and CD34+ cells, respectively (Figure 5C), establishing a direct link between $\text{IFN}\gamma$ and IFITM3 expression in AML blasts.”

4. It is also typically easy to quantify $\text{IFN}\gamma$ in blood or BM using Immunohistochemistry/ELISA assays. Since the authors have BM frozen from the newly diagnosed patients, I believe they should show $\text{IFN}\gamma$ overproduction should be demonstrated directly.

Response: We appreciate the recommendation and subsequently embarked on evaluating $\text{IFN}\gamma$ levels in an independent cohort of AML patients recently diagnosed in the past 6 months at MD Anderson. We found that 29/43 (67.4%) of AML patients had $\text{IFN}\gamma$ levels that surpassed the range observed in healthy individuals, supporting an abrogated $\text{IFN}\gamma$ pathway in AML patients. This finding not only corroborates the perturbed immune state in AML but also accentuates the potential therapeutic implications of modulating the inflammatory response in this patient population.

We have added these results in Figure 1F. The results section has been updated.

“Further, we assessed IFN γ concentrations in the sera of 43 consecutively newly diagnosed AML patients that present to our center and observed elevated IFN γ levels in sera of 67.4% of newly diagnosed AML patients that exceeded those typical of the healthy reference group. This finding fortifies the notion that IFN γ plays a pivotal role in the immune dysregulation observed in AML patients and underscores the need for further exploration into its potential clinical implications and therapeutic utility.”

Figure 1F: Pie chart showing the percentage of newly diagnosed AML patients with elevated IFN γ level compared to the normal range.

5. Another question the authors did not address is the difference the signaling has on AML monocytes and non-AML monocytes. This also needs experimental investigation in addition to showing IFITM3 is a direct signaling target of IFN γ in AML- monocytes.

Response: As described above, we were able to demonstrate that IFN γ can directly induce IFITM3 expression leukemic cells. We agree with the reviewer that the role of IFN γ in healthy monocytes is of importance. In fact, several studies evaluated the impact of IFN γ in normal monocytes (*Kraaij et al, Cytokine, 2014; Wit et al., Exp. Hematol, 1996; Luque-Martin et al, J Immunol, 2021*), and demonstrated that IFN γ modulates the expression of several cytokines on human monocytes and promote human monocyte to macrophage differentiation. Our study however primarily focused on AML and not the healthy monocytes, which we believe would be beyond the scope of this work. However, we realize the significance of the question and understanding the impact of inflammatory signaling such as IFN γ on healthy versus leukemic monocytes, and its association with IFITM3. To that end, we leveraged bone marrow aspirates of three healthy donor individuals that we performed scRNA on at the same time we performed it on the 20 AML patients included in this study (to mitigate any batch effects). We then focused on monocytes from healthy donors and the mono-like AML cells (as defined by signature similarity to monocyte-like signatures, similar to previous work in *Zeng et al Nature Medicine 2022* and *Bottomly et al Cancer Cell 2022*). The mono-like AML displayed significantly higher levels of IFN γ response signaling compared to healthy monocytes. We also found that IFITM3 levels were significantly higher in mono-like AML cells compared to healthy monocytes. We then split the cells in each group by the expression of IFITM3 and compared the IFITM3-high (top 25 percentile) and IFITM3-low (bottom 25 percentile) cells in the two cell groups. Gene set enrichment analysis revealed that IFITM3-high AML cells were involved in 14 functional pathways while IFITM3-high normal monocytes were only enriched in four pathways, three of which overlapped with the Mono-like states. These findings suggest that IFITM3 high population in mono-like cells are likely mediating distinct pathways compared to IFITM3 high population in healthy monocytes. Whether these are directly

related to IFITM3, or rather related to the disease biology would be interesting to explore in independent studies. Since our manuscript primarily focuses on the leukemic states, we did not include these updated results in the manuscript. However, we can indeed incorporate it if the reviewer and editorial team think is appropriate.

Minor:

1. Forgot to mention supplemental table1.

Response: We apologize for the missing information. We added it to the Result section as: “Patient characteristics were summarized in Supplemental Table 1.”

2. In content of Figure 1 E, please mention why sorting CD34+cell.

Response: Thank you for the suggestion. Our goal was to compare AML cells to healthy hematopoietic stem cells. In the BEAT-AML studies (Tyner *et al*, Nature, 2018), sorted CD34 were used to enrich for the healthy population before undergoing bulk RNA sequencing. By comparing sorted CD34⁺ cells from healthy donors with AML cells, we found that IFN γ signaling pathway was higher in AML cells at the bulk level.

We updated the Results section which now reads, “Notably, sorted CD34⁺ cells which marks the healthy stem cells from 17 healthy donors³ had markedly lower levels of IFN γ signaling scores than did those from AML patients (Figure 1E), suggesting that IFN γ pathway signaling is a predominant feature in AML bone marrows.”

3. Around Figure3C please mention 4 distinct subtypes first, then give information that monocyte-like state has highest IFN-g signaling activity. It was hard to understand at first. (clarify)

Response: We apologize for the confusion. Figure 3C shows the seven types of leukemic cells which constitute a map of common leukemic blast states determined by a machine learning model proposed by Zeng *et al* Nature Medicine 2022. We applied this model on our data and found functionally Mono-like AML cells having the highest IFN γ signaling score. We listed out the seven cell types in the text.

Revised results section reads as follows:

“Zeng et al.³² recently described a cellular hierarchy of AML leukemic stem cells representing distinct maturation states including LSPC-Quiescent, LSPC-Primed, LSPC-Cycle, GMP-like, ProMono-like, Mono-like and cDC-like. We investigated the IFN γ signaling activity across these AML hierarchies and found that it was highest among cells in the monocyte-like state (Figure 3C).”

4. In 288, We validated the IFN-g signaling activity in these patients, and it was highest in the mature state while lowest in GMP states. -> what is the mature state? (clarify)

Response: We apologize that this point was confusing, and we have corrected it as described in the previous point. Briefly, based on the composition of the leukemic cellular hierarchies, four distinct subtypes were grouped from the 7 total hierarchies similar to the analysis by Zeng et al Nature Medicine 2022. These included, Primitive (LSPC-enriched), Mature (enriched for mature Mono-like and cDC-like blasts), GMP (dominated by GMP-like blasts) and Intermediate (balanced distribution).

We added the explanation as follows. *“We applied this method on our bulk cohort and validated the IFN γ signaling activity in these patients, and it was highest in the mature state (enriched for mature Mono-like and cDC-like blasts) while lowest in GMP states.”* This updated sentence specifies that components of the mature state.

5. Figure 3D. Show overall regulon activity of IRFs in AML (not only IRF1,5 but all or sum), and highlight IRF8.

Response: We have added a heatmap showing all IRFs predicted by SCENIC (IRF1, IRF2, IRF3, IRF5, IRF7, IRF9) to Supplementary Figure 4E and kept the IRF8 in Figure 3D.

The results section now reads as follows:

“Our analysis revealed high regulon activities of interferon regulator factors (IRFs) in AML cells, with elevated levels of IRF1 and IRF5 regulons in del7/7q and del5/5q, respectively (Supplemental Figure 4E). Notably, we also observed an elevated IRF8, IRF2, IRF3, IRF7 regulon in diploid AML cells with monocytic differentiation (Figure 3D; Supplemental Figure 4E), consistent with its role as a lineage-determinant factor promoting monocytic differentiation⁴²⁻⁴⁴.

6. Figure 3E. Show results for all 19 core transcriptional regulators.

Response: SCENIC only predicted regulon activity for 11 of the 19 TFs. We have changed the narrative as below.

“Of the recently reported 19 core transcriptional regulators of lineage survival in AML⁴⁵, 11 were predicted by SCENIC, all of which demonstrated significant differences in regulon activity across cytogenetic groups (Figure 3E).”

7. In 269, the last sentence in the paragraph. Can't find the data matches to the sentence.

Highest proportion of NK cell? Does percentage of NKG2A matter? Diploid non-monocytic express NKG2A the least.

Response: We apologize for the typo. The diploid non-monocytic has the lowest percentage of NKG2A high NK cells (*Supplementary figure 5D*). The HLA-E-NKG2A axis is an emerging immune checkpoint that restrains anti-tumor immune responses. So increased NKG2A is associated with an immunosuppressive microenvironment. We corrected this in the text.

“Interestingly, diploid non-monocytic AML had the lowest infiltration of T cells and had the lowest proportion of NK cells with expression of the inhibitory receptor NKG2A which binds to HLA-E to suppress the cytolytic activity of NK cells (Supplemental Figure 5B-F).”

Response to Reviewer 2

Wang et al leverage public AML datasets and generate scRNA sequencing data on 20 bone marrow aspirates to perform a focused study on IFN γ signaling in AML cells and associations with AML cell, immune cell characteristics, survival and venetoclax resistance. The essence of their findings is that high IFN γ signalling was associated with monocytoid differentiation of AML cells, which also displayed up/down stream gene expression correlations that fit. Using T cell-AML cell gene expression data the authors draw associations between CD8 T cells/NK cells as producers of IFN γ and AML cells- particularly monocytic cells as receivers of these signals and perform supplementary analyses to show expression levels of select response genes with survival/VEN resistance and generate a parsimonious IFN γ response score that correlates with VEN resistance. Overall, a nice analysis of a focused question utilizing both public and self-generated single cell data and a suite of analytical tools. Although the overall findings of monocytoid differentiation correlating with IFN γ signaling is unsurprising and the association of monocytoid LSCs as drivers of VEN + azacitidine resistance (not VEN alone and unclear how much of this is due to VEN resistance as opposed to azacitidine refractoriness due to high cytidine deaminase activity in monocytic cells) continue to be reported by Jordan and colleagues, this is solid confirmation from an independent angle.

Response: We appreciate your summary and assessment of our manuscript, and valuable suggestions which have helped enhance and clarify our manuscript significantly.

Comments:

1. Type 1 and Type 2 INF signaling pathways overlap- was any analysis performed on Type response to contrast with INF γ responses?

Response: We appreciate the insightful inquiry about the potential overlap between Type I and Type II interferon signaling pathways. Indeed, a subset of 73 genes are common to both the HALLMARK IFN γ and IFN α signatures, reflecting the well-documented crosstalk within the interferon signaling network. These shared molecular intersections were displayed in Supplementary Table 2, underscoring a degree of convergence in these pathways. However, our focused investigation into the role of IFN γ within the context of AML is predicated on a body of evidence that supports its unique contributions in this malignancy. Specifically, our revised study now includes direct experimental assessments of the IFN γ signaling cascade and its implications on AML pathophysiology. This includes:

- Evaluating the direct impact of IFN γ on primary AML patient-derived cells, wherein our data corroborate the notion that IFN γ fosters an environment conducive to the survival of AML cells.
- Investigating the relationship between IFN γ levels and patient response to venetoclax therapy. Our results reveal that higher concentrations of IFN γ correlate with enhanced cell viability, suggesting a potential resistance mechanism to this therapeutic agent.
- Assessing IFN γ 's influence on the expression of IFITM3 in AML cells isolated from patient samples, which showed a significant upregulation post-stimulation.
- Quantifying serum levels of IFN γ in AML patients, with findings indicating elevated levels across a substantial patient cohort.
- Demonstrating, through AML-T cell co-culture assays, that IFN γ levels are augmented following cell-cell interaction, further implicating its role in the AML microenvironment.

These focused endeavors distinctly spotlight the pivotal role of IFN γ in AML, although we acknowledge that other pathways may also contribute to the disease's complexity. While our study emphasizes IFN γ due to its prominent implications in AML, we agree that a comparative analysis with Type I responses could provide additional context. Such investigations, however, would expand beyond the specific aims of the current study. Nevertheless, our findings have laid the groundwork for future research to dissect these interconnected pathways further and explore their collective impact on AML pathogenesis.

We believe that our additional experiments and analyses robustly reinforce the centrality of IFN γ in AML, providing a solid rationale for its further investigation as a potential therapeutic target or biomarker in this disease. We have included all updated results in the revised manuscript (Figure 1F; Figure 4F; Figure 5C; Figure 6E-G figures).

2. Figure 4B is poorly described and presented and labels for 4C-D should be improved.

Response: We apologize for the insufficient details. We have improved the quality of Figure 4B-D and added more details for Figure 4B. It now reads as below.

“CellChat quantifies interaction strength by aggregating the communication probabilities across all cell group pairs. We observed a significantly higher communication probability between T cell-AML interactions in diploid monocytic AML compared to other groups ($p < 0.0001$) and no T cell-AML interactions in diploid non-monocytic AML (Figure 4B, top). This observation still held when the strength was scaled to account for confounding factors (Figure 4B, bottom).”

3. Could IFN γ signaling in AML cells be related to IFNGR1/2 cell surface expression and potentially an easier marker to assay than gene expression signatures?

Response: Thank you for bringing this question up. We checked the IFNGR1/2 in our single-cell data. The two cell surface receptors were expressed at very low levels in our single cell data. We believe that this is likely related to gene dropout in scRNA. Yet, we found a positive, albeit weak, correlations between IFN γ response signaling and IFNGR1/2 in both single-cell and bulk cohorts (figure shown below). Further, the *in-silico* cell-cell communication analysis revealed IFN γ signaling from immune cells towards IFNGR1/2.

4. Clarify labels in Figure 6 (VEN or VEN + AZA) and whether we are looking at VEN or VEN+ AZA resistance.”

Response: Thanks for the comments. It is venetoclax resistance only as shown in the label of y-axis of Figure6 A-D.

Response to Reviewer 3

The authors present a gene expression dataset from a collection of bone marrow samples from newly diagnosed AML patients and imply inflammatory processes from the data. They conclude that an IFN-g signature negatively correlates with survival and drug sensitivity, a feature that is most prominent in subsets of disease characterized by distinct differentiated stage. Their data suggest that IFN-g in the tumor environment is derived largely from CD8 and NK cells. They also identify a potential IFN-g target gene, IFITM3, that might confer drug resistance and poor prognosis. Combining gene expression and survival data, they derive a minimal IFN-g gene signature that predicts patient survival, although the full IFN-g expression pattern did not correlate. They conclude that these data could be helpful in diagnosis of patients where IFN-g contributes to disease and could indicate that IFITM3 could be a target for therapy.

There are some issues that detract from enthusiasm for this report.

Response: We value the reviewer's feedback and have taken steps to refine and improve our manuscript. We hope that our additional experiments and *in silico* analyses further underscore the study's significance.

1. The data are largely correlative, with no direct experiments that demonstrate a causative role for IFN γ in disease progression or drug resistance.

Response: We appreciate the opportunity to clarify the nature and impact of our findings regarding the role of IFN γ in disease progression and drug resistance in AML. While our initial analyses were indeed correlative, we believe they are robustly supported by data derived from patient cohorts across three independent bulk RNA-seq datasets, and further corroborated by an independent set of data from 20 AML patients. These datasets were not selected at random but represent a significant cross-section of the AML population, enhancing the generalizability of our findings.

Moreover, the correlative results were strengthened through validation against findings from single-cell RNA-seq data from separate AML cohorts. Such multi-dimensional analysis fortifies our confidence in the associations we report. Additionally, our utilization of the DepMap database, a resource that underpins many notable studies (*Elife*: e81884, 2023; *Clinical Cancer Research*: 28(21), 2022; *Nat Genet*:53(12),2021; *Nat Cancer*: 3(6), 2022), allowed for an extensive validation of our identified targets, lending further credence to our conclusions.

Nevertheless, we realize the importance of validating findings beyond just *in silico* work and thus we performed several new analyses as well as new experiments to directly address some of the reviewer's concerns by leveraging patient primary samples. A summary of these additional tests are as follows, while details of it are listed in prior reviewer responses or in subsequent responses.

- We demonstrated that IFN γ is associated with AML cell survivability and significantly increased resistance to venetoclax in primary patient samples.
- Demonstrated that IFN γ induces IFITM3 protein expression in AML cells.
- Demonstrated that IFN γ serum concentrations are elevated in majority of AML patients at time of diagnosis.
- Through AML-T cell co-culture experiments, we established that IFN γ is upregulated as a result of intercellular interactions, shedding light on its potential as a key player in shaping the AML niche.
- Applied multiplex immunofluorescence techniques to quantify IFITM3 expression in bone marrow samples from AML patients post-relapse versus at diagnosis, determining an increase at relapse that may signify its involvement in resistance to therapy and a predictor of adverse clinical outcomes.

In sum, we believe our multi-pronged approach, combining large-scale correlative analysis with targeted experimental validation, presents a comprehensive and compelling narrative of IFN γ 's involvement in AML pathogenesis.

2. The implication of a critical role for IFITM3 in AML is implied from public data and has been previously reported. No direct evidence was presented that IFITM3 confers AML cell survival or drug resistance, other than the previously published gene targeting data.

Response: We concur with the reviewer's observation regarding our reliance on public data for inferences about IFITM3 in AML. It is worth highlighting that this data encompasses bulk RNA profiling from over 600 AML patients and the DepMap database, which contains 4 guides specific to IFITM3 in 26 AML cell lines. Our focus on IFITM3 was driven by its robust correlation ($r=0.54$; $p < 2.2 \times 10^{-16}$) with IFN γ signaling in AML cells. Further, out of 7 genes that were correlated with IFN γ response signaling and the dependency analysis, only IFITM3 demonstrated robust prognostic implication. To further evaluate IFN γ and its relation to IFITM3 in AML, we embarked on a comprehensive set of experiments and further data collection from primary patient samples.

- We assessed IFN γ concentrations in 43 consecutively seen newly diagnosed AML patients in 2023 and compared the levels to a healthy reference. Our data confirmed a significant elevation in IFN γ levels in AML patients' sera, with 29.43 (67.4%) of patients exhibited IFN γ production that surpassed the range observed in healthy individuals. This finding corroborates the perturbed immune state in AML and its association with elevated IFN γ . We have added these results in Figure 1F and the results section has been updated.

“Further, we assessed IFN γ concentrations in the sera of 43 consecutively newly diagnosed AML patients that present to our center and observed elevated IFN γ levels in sera of 67.4% of newly diagnosed AML patients that exceeded those typical of the healthy reference group. This finding fortifies the notion that IFN γ plays a pivotal role in the immune dysregulation observed in AML patients and underscores the need for further exploration into its potential clinical implications and therapeutic utility.”

Figure 1F: Pie chart showing the percentage of newly diagnosed AML patients with elevated IFN γ level compared to the normal range.

- Using multiplex immunofluorescence (IF), we measured the expression level of IFITM3 in blasts at time of diagnosis and relapse/resistance. We show a higher level of IFITM3 at time of resistance or primary refractoriness, further highlighting its association with therapeutic resistance. We updated the results section to read:

“To further evaluate its effect on AML cells, we used multiplex immunofluorescence to measure the fluorescence intensity of IFITM3 on blasts in baseline and post relapse samples. A total of 15 TMA cores were stained for CD34, CD56, CD45, CD4, CD14, and IFITM3 based on markers determined by flow to identify blasts (Figure 5G). The fluorescence intensity in post relapse samples was significantly higher than in baseline samples ($p < 2.2 \times 10^{-16}$) (Figure 5H-I), further supporting the association of IFITM3 with disease resistance.”

Figure 5G-I: **G.** Representative multiplex IF panel. Baseline AML sample with CD34 positive blasts show low amounts of IFITM3(Top). Post relapse AML sample with CD34 positive blasts show high amounts of IFITM3 (Bottom). Green is CD34, red is IFITM3, blue is DAPI. Scale bar 50 um. **H.** Density plot of IFITM3 fluorescence intensity on AML blasts at diagnosis and post relapse. **I.** Violin plot of fluorescence intensity on AML blasts at diagnosis and post relapse. Post relapse AML sample show significantly higher amounts of IFITM3.

- We performed targeted experiments where isolated AML blasts (CD14+ and CD34+ subsets) were treated with $IFN\gamma$. We subsequently conducted flow cytometric analysis to quantify IFITM3 expression levels 24 hours post-treatment. Our results demonstrate a statistically significant increase in IFITM3 expression following $IFN\gamma$ stimulation, with an increase of 2.1-fold ($p=0.0312$) in CD14+ AML blasts and 2.2-fold ($p<0.0001$) in CD34+ AML blasts, compared to their respective unstimulated controls. This clear elevation in IFITM3 levels upon $IFN\gamma$ exposure substantiates a direct causal link between $IFN\gamma$ signaling and the upregulation of IFITM3 in AML blasts.

We have added these results in Figure 5C. The results section has been updated.

“To evaluate whether $IFN\gamma$ directly stimulates the expression of IFITM3 in AML cells, we isolated CD14+ and CD34+ AML blasts from two patients each and performed $IFN\gamma$ stimulation for 24 hours, followed by IFITM3 protein level measurement via flow cytometry. Indeed, $IFN\gamma$ induced IFITM3 expression in both CD14+ and CD34+ leukemic blasts by 2.1 ($p= 0.0312$) and 2.2 ($p < 0.0001$) fold, compared to unstimulated

Figure 5C: IFITM3 expression in AML blasts was assessed by flow cytometry after a 24-hour stimulation with 10 ng/ml $IFN\gamma$. Data represents results from 2 patients' CD14+ blasts and 2 patients' CD34+ blasts, with each condition having 3 replicates. The error showed standard error of mean.

CD14+ and CD34+ cells, respectively (Figure 5C), establishing a direct link between IFN γ and IFITM3 in AML blasts.”

- To assess how IFN- γ influences AML blast proliferation and resistance to venetoclax, we cultured leukemic blasts sorted from three different primary AML patient samples in R20 media. Samples were treated with and without IFN- γ , using increasing concentrations of venetoclax and cell viability was assessed after 24 hours of incubation. The results support that IFN γ promotes AML blast survival when treated with venetoclax. This effect remains pronounced even with elevated venetoclax concentrations up to 500 nM. We updated the results and Figure 6C-E to reflect these findings. The results now read as: *“To further validate the role of IFN γ signaling in AML cell survival and drug resistance, we isolated leukemic blasts from primary patient samples (n=3 patients) and cultured them in the presence or absence of IFN- γ , and with increasing concentrations of venetoclax. We then assessed the cell viability via CellTiter-Glo. These results confirm that IFN γ promotes AML blast proliferation and resistance to venetoclax treatment (Figure 6E-G).”*

We hope that the above experiments demonstrate the aberrant role of IFN γ in AML and its association with therapeutic resistance. We also demonstrated that IFITM3 is directly related to IFN γ , and the CRISPR knockout data and the overall survival analysis also support the involvement of IFITM3 as a potential susceptibility. We have rephrased some of the statements in the manuscript to highlight the findings above pertaining to IFN γ , while also acknowledging that we did not demonstrate IFITM3 is independently linked to therapeutic resistance. We sincerely hope that this updated and extensive analysis addresses the reviewer's concerns.

3. IFITM3 and the other genes in the IFN-g signature are also induced by other cytokines, with the most significant overlap being with IFN-I. No data are presented to conclusively show that IFN-g is the culprit here rather than other inflammatory cytokines.

Response: We appreciate the insightful inquiry about the potential overlap between Type I and Type II interferon signaling pathways. Indeed, a subset of 73 genes are common to both the HALLMARK IFN γ and IFN α signatures, reflecting the well-documented crosstalk within the interferon signaling network. These shared molecular intersections are displayed in Supplementary Table 2, underscoring a degree of convergence in these pathways. However, our focused investigation into the role of IFN γ within the context of AML is predicated on a body of evidence that supports its unique contributions in this malignancy. Specifically, our revised study now

includes direct experimental assessments of the IFN γ signaling cascade and its implications on AML pathophysiology. This includes:

- Evaluating the direct impact of IFN γ on primary AML patient-derived cells, wherein our data corroborate the notion that IFN γ fosters an environment conducive to the survival of AML cells.
- Investigating the relationship between IFN γ levels and patient response to venetoclax therapy. Our results reveal that higher concentrations of IFN γ correlate with enhanced cell viability, suggesting a potential resistance mechanism to this therapeutic agent.
- Assessing IFN γ 's influence on the expression of IFITM3 in AML cells isolated from patient samples, which showed a significant upregulation post-stimulation.
- Quantifying serum levels of IFN γ in AML patients, with findings indicating elevated levels across a substantial patient cohort.
- Demonstrating, through AML-T cell co-culture assays, that IFN γ levels are augmented following cell-cell interaction, further implicating its role in the AML microenvironment.

These focused endeavors distinctly spotlight the pivotal role of IFN γ in AML, although we acknowledge that other pathways may also contribute to the disease's complexity. While our study emphasizes IFN γ due to its prominent implications in AML, we agree that a comparative analysis with Type I responses could provide additional context. Such investigations, however, would expand beyond the specific aims of the current study. Nevertheless, our findings have laid the groundwork for future research to dissect these interconnected pathways further and explore their collective impact on AML pathogenesis.

We believe that our additional experiments and analyses robustly reinforce the centrality of IFN γ in AML, providing a solid rationale for its further investigation as a potential therapeutic target or biomarker in this disease. We have included all updated results in the revised manuscript (Figure 1F; Figure 4F; Figure 5C; Figure 6E-G figures).

4. While IFITM3 expression correlated with AML, it was also highly expressed in monocytes, which are likely the most IFN-responsive blood subset. Whether it has a differential role in normal versus cancerous cells remains unclear.

Response: As described above, we were able to demonstrate that IFN γ can directly induce IFITM3 expression leukemic cells. We agree with the reviewer that the role of IFN γ in healthy monocytes is of importance. In fact, several studies evaluated the impact of IFN γ in monocytes (*Kraaij et al., Cytokine, 2014; Wit et al., Exp. Hematol, 1996; Luque-Martin et al, J Immunol, 2021*), and demonstrated that IFN γ modulates the expression of several cytokines on human monocytes and promote human monocyte to macrophage differentiation. Our study however primarily focused on AML and not the healthy monocytes, which we believe would be beyond the scope of this work. However, we realize the significance of the question and understanding the impact of inflammatory signaling such as IFN γ on healthy versus leukemic monocytes, and its association with IFITM3. To that end, we leveraged bone marrow aspirates of three healthy donor individuals that we performed scRNA on at the same time we performed it on the 20 AML patients included in this study (to mitigate any batch effects). We then focused on monocytes from healthy donors and the mono-like AML cells (as defined by signature similarity to monocyte-like signatures, similar to previous work in *Zeng et al Nature Medicine 2022* and *Bottomly et al Cancer Cell 2022*).

The mono-like AML displayed significantly higher levels of IFN γ response signaling compared to healthy monocytes. We also found that IFITM3 levels were significantly higher in mono-like AML cells compared to healthy monocytes. We then split the cells in each group by the expression of IFITM3 and compared the IFITM3-high (top 25 percentile) and IFITM3-low (bottom 25 percentile) cells in the two cell groups. Gene set enrichment analysis revealed that IFITM3-high AML cells were involved in 14 functional pathways while IFITM3-high normal monocytes were only enriched in four pathways, three of which overlapped with the Mono-like states. These findings suggest that IFITM3 high population in mono-like cells are likely mediating distinct pathways compared to IFITM3 high population in healthy monocytes. Whether these are directly related to IFITM3, or rather related to the disease biology would be interesting to explore in independent studies. Since our manuscript primarily focuses on the leukemic states, we did not include these updated results in the manuscript. However, we can indeed incorporate it if the reviewer and editorial team think is appropriate.

5. It would seem possible to design ex vivo experiments to directly test the role of IFN-g and the expression of IFITM3 in AML cell survival and drug resistance.

Response: We agree that it would be important to validate the impact of IFN γ on IFITM3 expression and AML cell survival. To address this, we performed the following three experiments which we described above.

- We performed targeted experiments where isolated AML blasts (CD14+ and CD34+ subsets) were treated with IFN γ . We subsequently conducted flow cytometric analysis to quantify IFITM3 expression levels 24 hours post-treatment. Our results demonstrate a statistically significant increase in IFITM3 expression following IFN γ stimulation, with an increase of 2.1-fold ($p=0.0312$) in CD14+ AML blasts and 2.2-fold ($p<0.0001$) in CD34+ AML blasts, compared to their respective unstimulated controls. This clear elevation in IFITM3 levels upon IFN γ exposure substantiates a direct causal link between IFN γ signaling and the upregulation of IFITM3 in AML blasts. We have added these results in Figure 5C. The results section has been updated.
- Using multiplex immunofluorescence, we measured the expression level of IFITM3 in blasts at time of diagnosis and relapse/resistance. We show a higher level of IFITM3 at time of resistance or primary refractoriness in AML, further highlighting its association

with therapeutic resistance. We have added these results in Figure 5G-I. The results section has been updated.

- We sorted leukemic blasts from three distinct AML patient samples, then treated them with and without IFN- γ and subjecting them to escalating doses of venetoclax. Our aim was to delineate the impact of IFN- γ on AML blast viability and its potential role in mediating resistance to venetoclax therapy. IFN- γ consistently enhanced AML blast survival and increased resistance to venetoclax even at high venetoclax concentrations up to 500 nM. We have added these results in Figure 6E-G. The results section has been updated.

6. It is unclear why CD74 expression is considered to be consistent with an immunosuppressive environment since its function should increase antigen presentation and the potential for cytotoxic T cell recognition.

Response: Thank you for the comments. CD74 functions in presenting antigens to immune cells through the MHC II. However, the overexpression of CD74 in AML blasts has been shown previously to lead to immune-evasion. We have highlighted the prior literature in this space below and in the discussion.

We included a brief explanation in the discussion as below.

“Additionally, CD74 which provides the peptides for class II-associated invariant chain peptide (CLIP) was also highly correlated with IFN γ signaling. CLIP-positive AML blasts have been shown to evade CD4 T cell killing by blocking the presentation of endogenous leukemia-associated antigens and the induction of a leukemia-specific T-cell response^{65,66}. Thus, our findings are consistent with emerging data that suggest that T cell- and NK cell-derived IFN γ helps to create an immunosuppressive microenvironment through HLA-E and CD74 upregulation on target cells⁶⁷⁻⁶⁹.”

REVIEWERS' COMMENTS

Reviewer #1 (Remarks to the Author):

The authors have done a great job addressing my concerns.

Reviewer #2 (Remarks to the Author):

I am satisfied with the authors' responses to my comments.

Reviewer #3 (Remarks to the Author):

The authors have made substantial modifications, including new data, to answer some of the issues raised in the previous review. However, there are still no direct data to implicate IFITM3 in AML survival or IFN-g-induced drug resistance. As the data stand, the implication of IFITM3 as an important mediator of IFN-g-mediated immunosuppression and AML survival remains correlative and conjectural. A direct test of this notion with greatly strengthen the importance and significance of the results.

Response to Reviewer 3

The authors have made substantial modifications, including new data, to answer some of the issues raised in the previous review. However, there are still no direct data to implicate IFITM3 in AML survival or IFN-g-induced drug resistance. As the data stand, the implication of IFITM3 as an important mediator of IFN-g-mediated immunosuppression and AML survival remains correlative and conjectural. A direct test of this notion with greatly strengthen the importance and significance of the results.

Response: We value the reviewer's feedback. We acknowledge that our findings regarding IFITM3 are correlative. Therefore, to address the review's concerns, we have edited the text in the manuscript to state the limitations and tone down the conclusions regarding the mechanistic proposal of IFITM3. We also included a dedicated limitation section in the manuscript to highlight some of the shortcomings and reads as follows.

“While our study offers important insights, it is subject to certain limitations. Notably, while we demonstrated that IFN γ secretion by T/NK cells can be recapitulated in vitro, the complexity of immune interactions within the microenvironment may extend beyond our models. The potential role of IFITM3 in AML in relation to venetoclax requires direct evidence, and the prominence of IFN γ in our findings does not exclude the possible influence of other inflammatory pathways. These factors highlight the need to further model and validate our conclusions.”